# Thermodynamically consistent machine learning model for excess Gibbs energy

Marco Hoffmann [1,4], Thomas Specht [1,4], Quirin Göttl[2], Jakob Burger[2], Stephan Mandt [3], Hans Hasse[1] & Fabian Jirasek [1] ✉

The excess Gibbs energy plays a central role in chemical engineering and chemistry, providing a basis for modeling thermodynamic properties of liquid mixtures. Predicting the excess Gibbs energy of multi-component mixtures solely from molecular structures is a long-standing challenge. We address this challenge with HANNA, a flexible machine learning model for excess Gibbs energy that integrates physical laws as hard constraints, guaranteeing thermodynamically consistent predictions. HANNA is trained on experimental data for vapor-liquid equilibria, liquid-liquid equilibria, activity coefficients at infinite dilution, and excess enthalpies in binary mixtures. The end-to-end training on liquid-liquid equilibrium data is facilitated by a surrogate solver. A geometric projection method enables robust extrapolations to multi-component mixtures. We demonstrate that HANNA delivers accurate predictions, while providing a substantially broader domain of applicability than state-of-the-art benchmark methods. The trained model and corresponding code are openly available, and an interactive interface is provided on our website, MLPROP.

Accurate knowledge of the thermodynamic properties of liquid mixtures is crucial in chemistry, chemical engineering, and environmental science. Among these properties, the molar excess Gibbs energy, $g^E$, plays a central role: it provides the foundation for deriving many other thermodynamic quantities through established relations. In particular, $g^E$ serves as the basis for determining the activity coefficients of the mixture's components.

Activity coefficients, in turn, govern phase equilibria, such as vapor–liquid (VLE), liquid–liquid (LLE), and solid–liquid (SLE) equilibria, and are key to describing chemical equilibria. They are also used in modeling reaction kinetics and transport phenomena. This makes $g^E$ a central property in model-based process design and optimization. However, $g^E$ and the derived activity coefficients cannot be measured directly but must be inferred from phase equilibrium data, and the corresponding experiments are costly and time-consuming.

Given the many thousands of technically relevant components that can form an enormous number of mixtures, and the fact that $g^E$ depends on the state point (especially temperature and composition in liquid mixtures), only a tiny fraction of all relevant mixtures can be studied experimentally. This makes the development of predictive models for $g^E$ a central task in thermodynamics. In this context, we distinguish between different classes of predictions, namely predictions for unstudied state points, that is, temperatures and/or compositions of a mixture for which data exist at other conditions; unstudied systems, meaning combinations of components for which data exist in other systems; and unstudied components, referring to entirely new substances for which no data exist.

All common $g^E$ models are based on the concept of pair interactions. The most widely used $g^E$ models are based on the local-composition theory, such as NRTL[1] and UNIQUAC[2], in which the molecular interactions are described by pair-interaction parameters between components, which are usually fitted to experimental data for each binary system of interest. These models enable predictions for unstudied temperatures and compositions, as well as for unstudied multi-component systems, provided data exist to fit the model parameters of all constituent binary subsystems. However, these models

[1]Laboratory of Engineering Thermodynamics, RPTU University Kaiserslautern-Landau, Kaiserslautern, Germany. [2]Laboratory of Chemical Process Engineering, Technical University of Munich, Munich, Germany. [3]Department of Computer Science & Statistics, University of California, Irvine, CA, USA. [4]These authors contributed equally: Marco Hoffmann, Thomas Specht. ✉e-mail: fabian.jirasek@rptu.de

do not allow predictions for unstudied components, binary systems, and multi-component mixtures comprising unstudied binary subsystems. Furthermore, these models suffer from limited flexibility, which often hinders the simultaneous description of VLE and LLE with a single parameterization[3,4].

To address the lacking capability of predictions to unstudied systems and components, group-contribution (GC) $g^E$ models have been developed, with the different variants of UNIFAC being the most successful ones[5–8]. UNIFAC decomposes components into structural groups and utilizes parameters that describe pair interactions between these groups, significantly reducing the number of parameters and enabling extrapolations to unstudied systems and components. However, models of the UNIFAC family are limited to mixtures for which all involved structural groups (and their interactions) have previously been parametrized. Because the original UNIFAC model[5] is unable to describe both VLE and LLE with high accuracy, a special UNIFAC parameterization for LLE has been introduced[9]. Additionally, for describing complex systems, such as those containing ionic liquids[10,11] or formaldehyde[12,13], tailored groups had to be introduced, compromising the GC nature of UNIFAC and the generality of the approach. Despite the described shortcomings, the different UNIFAC variants have long been and remain the standard $g^E$ models used for the predictive modeling of phase equilibria across industry and academia, when no data for fitting parameters for models such as UNIQUAC and NRTL are available.

An alternative to the GC models of the UNIFAC family are the $g^E$ models based on quantum-chemical density functional theory (DFT) calculations combined with the conductor-like screening model (COSMO)[14]. Different variants of such frameworks have been published, most notably COSMO-RS[15] and COSMO-SAC[16]. While these models avoid group-interaction parameter tables and can, in principle, be applied to any system, they often require expensive calculations, and the results can depend on the chosen DFT level and COSMO parametrization[17–20]. For systems where both approaches can be applied, UNIFAC is often reported to be more accurate than COSMO frameworks[17,21,22].

Recently, several authors used machine learning (ML) methods to further extend the scope of physics-based $g^E$ models by predicting their pair-interaction parameters. This was, e.g., done for UNIQUAC[23], NRTL[24], and UNIFAC[25,26]. These new hybrid methods rely on the established $g^E$ models and are therefore intrinsically thermodynamically consistent. However, these models also inherit all the limitations of the underlying $g^E$ models, e.g., the problems of simultaneously describing VLE and LLE, or the restriction to a predefined list of structural groups[25,26] or components[23].

Furthermore, several ML models for the prediction of activity coefficients have been proposed that are not based on the equations of any of the physics-based $g^E$ models. Medina et al. presented graph neural network (GNN) architectures for the prediction of infinite-dilution activity coefficients at constant temperature (GNN-IAC)[27], and with temperature dependence (GH-GNN)[28], using the Gibbs-Helmholtz equation. Qin et al.[29] have developed a GNN (SolvGNN) for the prediction of composition-dependent, isothermal activity coefficients in binary and ternary mixtures. Based on the SolvGNN architecture, Rittig et al. have built the Gibbs-Duhem-informed GNN (GDI-GNN)[30] and the excess Gibbs free energy GNN (GE-GNN)[31], which include soft constraints and hard constraints on the Gibbs-Duhem equation, respectively.

In recent work, we have introduced HANNA[32], a model that combines the flexibility of artificial neural networks with explicit thermodynamic knowledge hard-coded into the network architecture. In its first version, HANNA was trained on binary activity coefficients derived from experimental VLE data, as well as experimental activity coefficients at infinite dilution (ACI). HANNA is the first model to predict temperature- and composition-dependent activity coefficients without being based on an underlying physics-based $g^E$ model, while still

strictly complying with all relevant thermodynamic consistency criteria. These comprise the consistency of the activity coefficients with the Gibbs-Duhem equation, permutation-equivariance (a permutation in the component input order must be equally reflected in the order of the activity coefficients in the output; their values must not change), their correct behavior at infinite dilution (for $x_i \to 1$, $\gamma_i \to 1$), and consistency for pseudo-mixtures (identical components in the mixture must reduce to one pure component).

In this work, a generalization of HANNA is presented that is flexible enough for the holistic description of different phase equilibria (namely VLE and LLE), enables predictions for multi-component mixtures comprising any number of components, and thereby still complies with all thermodynamic constraints. To achieve this, we significantly improve the training process, which now additionally includes end-to-end training on experimental LLE and excess enthalpy data. Moreover, we implement a geometric projection method for extrapolation to any multi-component mixture. The only input required for HANNA is the molecular structure of the components and the considered state point (temperature and composition). We provide the trained HANNA model to the community in a fully open-source format and also make it available through an intuitive user interface on our website MLPROP.

Figure 1 gives a high-level overview of HANNA and its training procedure. The network structure of HANNA includes permutation-invariant operations, ensuring that predictions are independent of the order of components in the input, as well as operations that guarantee consistency in the limiting cases of binary subsystems and pure components. To generalize to multi-component mixtures, we introduce a geometric projection method[33] that maps the $g^E_{ij}$ of the binary subsystems into the multi-component space. Because this projection introduces no additional parameters, it is sufficient to train the model only on data from binary mixtures. Details on the HANNA architecture and the projection are given in the section "HANNA architecture".

HANNA is trained end-to-end on extensive experimental datasets of binary mixtures, covering activity coefficients (normalized after Raoult's law) derived from VLE data for which temperature, pressure, and the composition of the liquid and vapor phase are given (TPXY data) and activity coefficients at infinite dilution (ACI data). We also include VLE data for which the vapor-phase composition is unavailable (TPX data) in the training, with total pressure as the training objective. Additionally, we train to phase compositions from LLE data and excess enthalpies (HE data). All data was taken from the Dortmund Data Bank (DDB)[34]. The complete dataset covers 824,481 data points for 46,543 unique systems of 4114 unique components with a large chemical diversity, also including ionic liquids. Details on the dataset can be found in the section "Data".

To include LLE data into the training process, we develop a new training strategy using a surrogate solver that circumvents the necessity for iteratively solving non-linear equations during training[35]. To the best of our knowledge, this is the first time that experimental phase compositions from LLE data are used in the end-to-end training of an ML-based $g^E$ model. Furthermore, to train HANNA on HE data, we leverage the temperature derivative of the predicted excess Gibbs energy, which is easily available through automatic differentiation. We also introduce a physically inspired "Gibbs loss" that enhances the recognition and description of LLE based on the stability criterion[35]. Additionally, a Lipschitz regularization strategy is employed, increasing HANNA's robustness outside the training regime by enforcing smoothness of the predicted $g^E$. Details on the model training are given in the section "Training of HANNA".

In the following, we demonstrate the performance of HANNA for predicting activity coefficients derived from VLE (TPXY data), ACI data, and LLE phase compositions in binary and ternary mixtures. All results were obtained exclusively on unseen test systems withheld during the training of HANNA.

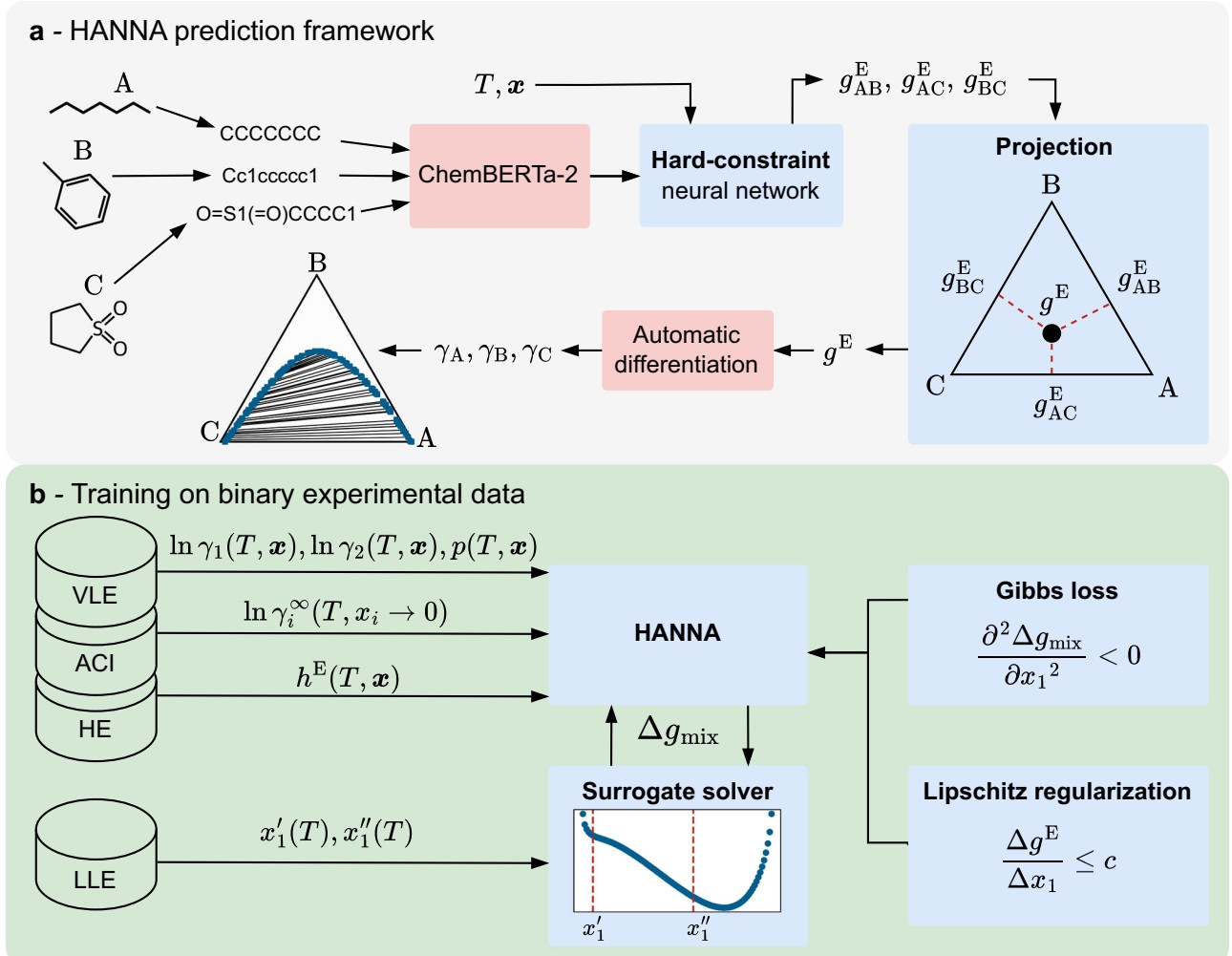

**Fig. 1 | Overview of the HANNA prediction framework and its training procedure. a** As input, HANNA requires the molecular structure of all components in the SMILES[62] notation, the composition of the mixture $\boldsymbol{x}$, and the temperature $T$. In the first step, molecular embeddings are generated from the SMILES using the transformer model ChemBERTa-2[43]. Subsequently, a hard-constraint neural network that strictly obeys all thermodynamic constraints predicts $g_{ij}^{E}$ of all binary subsystems, from which $g^E$ in the multi-component mixture is calculated via a geometric projection[33]. Thermodynamically consistent activity coefficients $\gamma_i$ are obtained by automatic differentiation of $g^E$ and then used to predict phase equilibria using the convex envelope method (CEM)[37–39]. **b** HANNA is trained on experimental vapor–liquid equilibrium (VLE), liquid–liquid equilibrium (LLE), activity coefficients at infinite dilution (ACI), and excess enthalpy (HE) data of binary mixtures. End-to-end training on the experimental LLE data is enabled by a new surrogate solver using the Gibbs energy of mixing $\Delta g_{mix}$. An additional Gibbs loss incentivizes the model to produce negative second derivatives of $\Delta g_{mix}$ in the LLE regime, and a Lipschitz regularization enforces smoother $g^E$ functions.

## Results

### Comparison to modified UNIFAC (Dortmund)

Figure 2 compares HANNA's predictive performance for binary mixtures against that of modified UNIFAC (Dortmund)[7,8], hereafter referred to as mod. UNIFAC. We use mod. UNIFAC is the primary benchmark in this work, because it is the most established predictive $g^E$ model in industry and academia.

The results shown in the boxplots of Fig. 2a demonstrate that HANNA has a higher prediction accuracy than mod. UNIFAC on the shared mod. UNIFAC horizon (i.e., all systems to which mod. UNIFAC could be applied) for all studied data types: activity coefficients from vapor–liquid equilibrium data (TPXY, top), activity coefficients at infinite dilution (ACI, middle), and phase compositions in liquid–liquid equilibrium (LLE, bottom). We assess the performance using the system-specific mean absolute error MAE$_{sys}$, cf. section "Evaluation of HANNA". The most significant difference is found for the ACI data, where the median of MAE$_{sys}$ is reduced from 0.18 (mod. UNIFAC) to 0.11 (HANNA). This substantial improvement could be related to a deficiency of UNIQUAC (on which UNIFAC is based) to correctly predict infinite dilution activity coefficients in strongly non-ideal systems[36].

Beyond the mod. UNIFAC horizon, the performance of HANNA slightly declines for the VLE and LLE data (cf. full horizon), which could be attributed to systems with more complex components containing rarely occurring groups that mod. UNIFAC fails to predict entirely because not enough data were available to regress the required parameters. Importantly, for the LLE data, error scores could only be calculated if the respective model predicts a miscibility gap at the given state point. Therefore, the number of systems for which predictions are obtained differs from the number of systems within the LLE dataset that are theoretically feasible with the models. Thus, for a fair comparison, the mod. UNIFAC horizon of the LLE data comprises only systems that mod. UNIFAC can model and for which both mod. UNIFAC and HANNA correctly predict a phase split. On the mod. UNIFAC horizon, mod. UNIFAC correctly identifies miscibility gaps for 74% of the LLE systems, whereas HANNA achieves 89% accuracy. On the full horizon, HANNA correctly identifies miscibility gaps for 83% of the LLE systems.

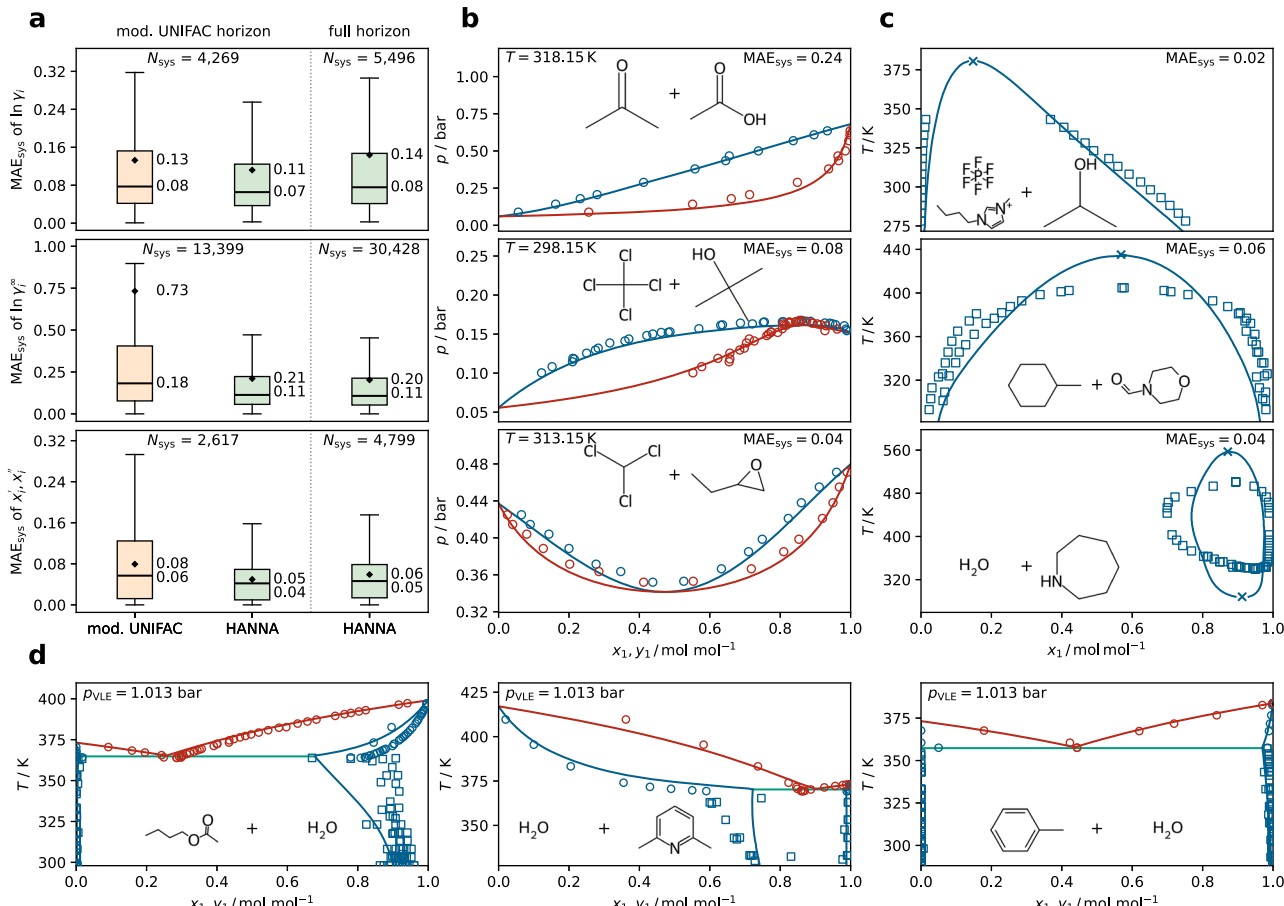

**Fig. 2 | Predictions for binary mixtures with HANNA. a** Boxplots comparing the performance of HANNA for predicting activity coefficients in binary VLE (top), ACI (middle), and LLE phase compositions (bottom) with that of mod. UNIFAC in terms of the system-wise mean absolute error MAE$_{sys}$ in $\ln \gamma_i$ (VLE, ACI) or phase compositions $x_i'$ and $x_i''$ (LLE). For a fair comparison, HANNA was also evaluated only on those systems for which mod. UNIFAC is applicable (mod. UNIFAC horizon). $N_{sys}$ denotes the number of test systems within the respective horizon and data type. The boxes represent interquartile ranges, and the whiskers are 1.5 times the interquartile range. Diamonds mark the mean, horizontal lines the median of the MAE$_{sys}$ values. **b** Predictions of HANNA for three isothermal VLE plotted against experimental data. Open symbols denote experimental data, and lines are predictions by HANNA. Blue indicates the liquid phase; red indicates the vapor

phase. The molecular structures of the components and the respective MAE$_{sys}$ are depicted in the plots; the left molecule corresponds to component 1.
**c** Predictions of HANNA for three LLE. Open squares indicate experimental data, lines represent predictions by HANNA. Predicted upper and lower critical solution temperatures are marked with an "x". **d** Predictions of HANNA for three isobaric heteroazeotropes. Open blue and red circles indicate the experimental liquid and vapor phases, respectively, from VLE data. Open blue squares mark the experimental phase compositions from LLE data; the green line is the vapor–liquid–liquid equilibrium predicted by HANNA. All results shown here are for systems from the test set that were not used for the training of HANNA. Source data are provided as a Source data file.

For the ACI data, the full horizon we evaluate HANNA on contains more than twice as many systems as the mod. UNIFAC horizon. The largest share of the additional systems contains at least one ionic liquid (IL). As only very few groups to model ILs have been parameterized within mod. UNIFAC, it has a very limited scope for systems containing ILs, whereas HANNA has no such restrictions. In Supplementary Fig. 10, we specifically evaluate the performance of HANNA for predicting activity coefficients in IL systems and compare it to that of mod. UNIFAC for the few IL systems for which mod. UNIFAC can be used. The results show significantly higher prediction accuracy for HANNA both on the mod. UNIFAC horizon and the full horizon.

Three typical examples for predictions of binary VLE diagrams with HANNA are shown in Fig. 2b. The predictions were obtained using extended Raoult's law with vapor pressures calculated from the Antoine parameters of the DDB[34]. For all systems, HANNA shows good agreement with the experimental data. HANNA correctly describes nearly ideal mixtures (top), mixtures with strong negative (middle), and mixtures with strong positive (bottom) deviations from Raoult's law.

Figure 2c demonstrates that HANNA can also predict different types of LLE phase diagrams. For the predictions, we assume a pressure-independent LLE. As for the ACI data, HANNA also yields good predictions for LLE involving ILs (top), and even qualitatively predicts an island-type miscibility gap with upper and lower critical solution temperatures (bottom).

In Fig. 2d, predictions of HANNA for three heteroazeotropic systems are shown. Both VLE and LLE are predicted well, a task that established $g^E$ models, such as UNIQUAC or NRTL, often only achieve with multiple parameter sets, fitted for different temperature ranges. The results demonstrate HANNA's applicability across a wide temperature range with a single parameterization. The phase diagrams in Fig. 2c, d were obtained using the convex envelope method (CEM)[37–39], which enables a seamless calculation of different types of phase equilibria, considering the global behavior of the studied system, and thus avoiding individual phase equilibrium calculations and the problems associated with this. In Supplementary Fig. 8, we evaluate the predictions of mod. UNIFAC on the systems of Fig. 2b–d that fall within the mod. UNIFAC horizon.

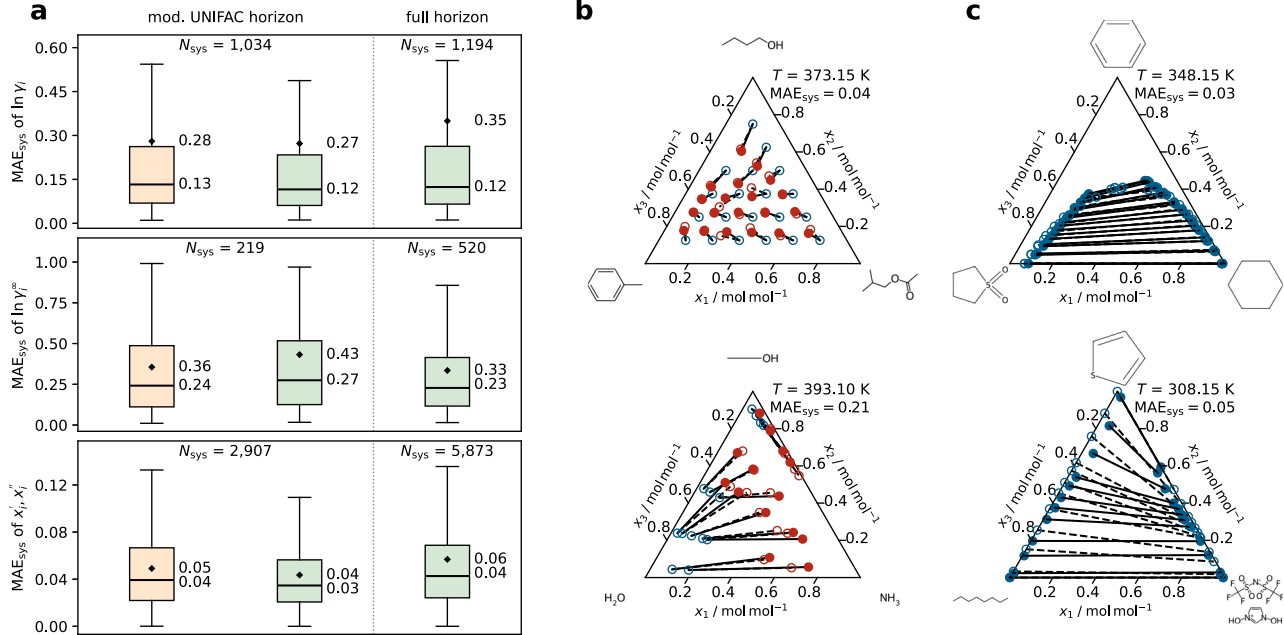

**Fig. 3 | Predictions for ternary mixtures with HANNA. a** Boxplots comparing the performance of HANNA for predicting activity coefficients in ternary VLE (top), ACI (middle), and LLE phase compositions (bottom) with that of mod. UNIFAC in terms of the system-wise mean absolute error, $MAE_{sys}$ in $\ln \gamma_i$ (VLE, ACI) or phase compositions $x_i'$ and $x_i''$ (LLE). For a fair comparison, HANNA was also evaluated only on those systems for which mod. UNIFAC is applicable (mod. UNIFAC horizon). $N_{sys}$ denotes the number of test systems within the respective horizon and data type. The boxes represent interquartile ranges, and the whiskers are 1.5 times the interquartile range. Diamonds mark the mean, horizontal lines the median of the

MAE$_{sys}$ values. **b** Predictions of HANNA for two isothermal VLE. Open blue and red symbols mark the experimental liquid and vapor phase compositions, respectively; filled red symbols are the predicted vapor phase compositions. The molecular structures of the components and the respective $MAE_{sys}$ are also depicted. **c** Predictions of HANNA for three isothermal LLE. Open symbols and dashed tie lines represent the experimental data; filled symbols and solid lines represent predictions by HANNA. All results shown here are for systems from the test set that were not used for HANNA training. Source data are provided as a Source data file.

In Supplementary Figs. 6 and 7, we compare HANNA and mod. UNIFAC for the prediction of excess enthalpies (HE data) and total pressures in VLE (TPX data), respectively. For both data types, HANNA yields more accurate predictions than mod. UNIFAC. Additionally, we also compare HANNA to mod. UNIFAC 2.0[26], an ML-enhanced version of mod. UNIFAC that exceeds the original model in scope due to a completed interaction parameter matrix, and to UNIFAC-LLE[9], a version of UNIFAC that was specifically parameterized to describe LLE data between 283.15 and 313.15 K. The comparisons to mod. UNIFAC 2.0 and UNIFAC-LLE for binary mixtures are presented in Supplementary Figs. 11 and 14, respectively.

Figure 3 evaluates HANNA's performance on unseen ternary mixtures. Similar to the results for the binary mixtures, Fig. 3a depicts boxplots of the system-wise MAE for the predictions of activity coefficients in ternary VLE (TPXY, top), infinite dilution activity coefficients in mixed solvents (ACI, middle), and LLE (bottom), and compares the performance of HANNA to that of mod. UNIFAC.

The results in Fig. 3a show higher or equal accuracy of HANNA for the ternary VLE (TPXY) and LLE data compared to mod. UNIFAC within the shared horizon, and a slight decline in the accuracy of HANNA on the full horizon. For the ternary ACI data, we observe slightly higher mean and median values of the $MAE_{sys}$ for HANNA than for mod. UNIFAC. Interestingly, the HANNA scores for ternary ACI data improve when going to the full horizon. On the mod. UNIFAC horizon, mod. UNIFAC correctly identifies the miscibility gaps for 77% of the ternary LLE systems, whereas HANNA achieves 98% accuracy. On the full horizon, HANNA correctly identifies the miscibility gaps for 92% of the systems.

Figure 3b, c shows HANNA predictions for ternary systems exhibiting a VLE or LLE, respectively, and compares them to experimental

data. For the LLE prediction, we again neglect the pressure dependence of $g^E$. Good agreement with the experimental data is found for all four systems. While the results in Fig. 3c cover ternary LLE with one (top) and two (bottom) binary subsystems that exhibit a miscibility gap, LLE examples for systems with complete miscibility of the binary subsystems (island-type LLE behavior) and miscibility gaps in all three binary subsystems (leading to a three-phase LLLE) are given in Supplementary Fig. 5. HANNA is also able to predict these correctly, at least qualitatively. Again, we have also compared HANNA against mod. UNIFAC 2.0 and UNIFAC-LLE, which yielded results similar to those against mod. UNIFAC shown here, cf. Supplementary Figs. 12 and 14, respectively.

Comparing HANNA's performance on binary and ternary mixtures reveals how different extrapolation strategies influence accuracy across VLE, ACI, and LLE data (cf. panels a in Figs. 2 and 3). For the ternary VLE data, the scores from HANNA and mod. UNIFAC become slightly worse by the same margin (their median $MAE_{sys}$ increases by about 0.05), compared to the respective binary scores. We observe a similar pattern for the LLE data. For the ACI data, we observe a significant deterioration in prediction accuracy when moving from binary to ternary mixtures for HANNA (the median $MAE_{sys}$ increases by 0.16).

The extrapolation from binary to multi-component data in mod. UNIFAC is based on the physical lattice model, which uses the pair-interaction concept. In contrast, HANNA uses a purely geometric projection method. To determine whether the projection method is responsible for the deterioration in HANNA for ternary ACI data, we conducted the following test: for all systems in the ternary VLE (TPXY) and ACI datasets, we predicted the corresponding activity coefficient curves for all binary subsystems at the respective temperatures using HANNA. We then fitted two component-specific physics-based $g^E$

**Table 1 | Comparison of HANNA with different ML models from the literature**

| | Model | TPXY | | TPX | | ACI | | LLE | |
|---|---|---|---|---|---|---|---|---|---|
| | | Mean | Median | Mean | Median | Mean | Median | Mean | Median |
| Binary | HANNA | **0.165** | **0.066** | **0.151** | **0.050** | **0.229** | **0.141** | **0.032** | 0.011 |
| | GE-GNN | 0.227 | 0.108 | 0.272 | 0.091 | 0.438 | 0.296 | 0.051 | 0.012 |
| | GDI-GNN | 0.225 | 0.109 | 0.278 | 0.090 | 0.433 | 0.288 | 0.046 | **0.009** |
| | SolvGNN | 0.225 | 0.109 | 0.280 | 0.091 | 0.432 | 0.293 | –[a] | –[a] |
| | GNN-IAC | –[b] | –[b] | –[b] | –[b] | 0.372 | 0.150 | –[b] | –[b] |
| Ternary | HANNA | **0.646** | **0.143** | **0.310** | **0.141** | **0.429** | **0.257** | – | – |
| | SolvGNN | 0.689 | 0.189 | 0.667 | 0.328 | 0.841 | 0.758 | –[a] | –[a] |

We evaluate all models (including HANNA) only on isothermal data at 298 K and exclude all systems containing ionic liquids from the comparison. Because the architecture of GNN-IAC is not suited to processing water, systems that contain water are omitted from the binary ACI comparison. The mean and median of the MAE$_{sys}$ for each data type are given. The best scores for each data type are bold. The results for HANNA are for test data; the other models were used as published. Source data are provided as a Source data file.
[a]Due to large oscillations in the predicted activity coefficients over the concentration, SolvGNN could not be evaluated on LLE data.
[b]GNN-IAC can only predict activity coefficients at infinite dilution.

models, namely UNIQUAC (the parent model of UNIFAC) and NRTL, to the predicted activity coefficients and used the resulting parameters to predict activity coefficients in the ternary systems. This procedure allows comparison only of the physics-based model extrapolations (UNIQUAC, NRTL) with the geometric extrapolation of HANNA, using the same binary data. The results of this comparison, which are shown in Supplementary Figs. 15 and 16, indicate a similar or even slightly better accuracy using the geometric projection compared to the extrapolation of the physics-based models. Hence, the geometric Muggianu projection method in the HANNA architecture generally works reasonably well. This is also supported by Supplementary Fig. 9, where we provide a comparison of HANNA and mod. UNIFAC on quaternary VLE and LLE data, showing comparable accuracy of both methods. Nonetheless, this does not exclude that further improvements can be made on the geometric extrapolation, for which different suggestions exist in the literature[40,41].

The shown results demonstrate that HANNA is a strong competitor to mod. UNIFAC, the most established model for predicting activity coefficients in industry and academia, both for binary and multi-component mixtures. This is especially remarkable because we have evaluated HANNA on data from the DDB, the largest factual database for thermodynamic properties, which is managed essentially by the same group that has revised mod. UNIFAC over several decades. Consequently, we must assume that mod. UNIFAC has been fitted to a significant part of our test data (although the concrete training data for mod. UNIFAC has not been fully disclosed). Similar arguments hold for mod. UNIFAC 2.0 and UNIFAC-LLE. In contrast, the results of HANNA have been obtained on test data, strictly withholding all available data types for the test systems during model training. Hence, most comparisons we have shown here are biased in favor of mod. UNIFAC. In addition to its strong predictive performance, HANNA offers a much broader applicability than mod. UNIFAC, because it can, in principle, be applied to any mixture for which the molecular structures of the components are known; remaining limitations are discussed in the section "Limitations" below.

## Comparison to other machine learning models

In the following, we also compare HANNA against different ML models from the literature. We thereby consider only ML models that, like HANNA, are not based on the equation framework of a physics-based $g^E$ model, require only the SMILES of the components as input, and for which the model parameters are disclosed or can be reproduced with the available code. Specifically, we have selected GE-GNN[31], GDI-GNN[30], SolvGNN[29], and GNN-IAC[27] as additional benchmarks. Compared to HANNA, all these models have limitations in their applicability: they are only applicable to binary systems (GE-GNN, GDI-GNN, GNN-IAC), to the

state of infinite dilution (GNN-IAC), or to predictions at $T$ = 298 K (all). Furthermore, they lack strict consistency with the Gibbs-Duhem equation (GDI-GNN, SolvGNN) or in the limiting case of pure components (GE-GNN, GDI-GNN, SolvGNN), whereas HANNA complies with all relevant constraints. Supplementary Table 2 summarizes the ranges of applicability of these models and the consistency criteria they comply with.

For a fair comparison, we evaluate HANNA and the other ML models on common subsets of our data for which predictions are feasible with all models. While the results of HANNA on these subsets are true predictions (since the respective systems were not part of the training or validation sets), we cannot rule out that some of them were included in the training sets of the literature models. We note that GE-GNN, GDI-GNN, and SolvGNN have not been trained on experimental data but on synthetic data generated with COSMO-RS[15].

The results of the comparison are summarized in Table 1. The scores of GE-GNN, GDI-GNN, and SolvGNN on the binary TPXY, TPX, and ACI data are almost identical, which could be expected, as all three models rely on the same GNN architecture and training dataset and only differ in the final step of the prediction of the activity coefficients and/or the regularization during training. In fact, the authors arrived at a similar conclusion when comparing GE-GNN against GDI-GNN and SolvGNN on their extrapolation test dataset[31]. For the TPXY, TPX, and ACI data, HANNA shows significantly lower scores (higher accuracy) for both binary and ternary systems. On the LLE subset, HANNA achieves the lowest mean MAE$_{sys}$, suggesting the fewest outliers, while no significant difference in the median MAE$_{sys}$ between the models is observed. Overall, HANNA shows the best performance, with the highest prediction accuracy in almost all cases and a significantly broader applicability than all benchmark models, cf. Supplementary Table 2.

## Limitations

We have demonstrated that HANNA provides accurate predictions for activity coefficients in binary and ternary mixtures, and that phase equilibria calculated from these predictions agree well with experimental data. The training data used here consists of VLE (TPXY and TPX), ACI, LLE, and HE data, with 95% in a temperature range between 273 and 428 K. Similar to other $g^E$ models, HANNA can also be used to predict activity coefficients for phase equilibria with supercritical components (using Henry's law) or solid–liquid equilibria (SLE). However, HANNA's performance on these additional data types and outside the described temperature range is yet to be evaluated. Furthermore, HANNA, like most $g^E$ models, neglects the pressure dependence of the liquid phase activity coefficients. While this is a valid assumption for most engineering applications, it may break down at very high pressures.

HANNA was trained on a large, diverse dataset and has learned to accurately predict activity coefficients for mixtures of components from many classes, including both small and large molecular components and components containing exotic chemical groups. Weak electrolytes that may dissociate partially and ionic liquids were also included in the training set and are therefore within the current scope of HANNA. However, HANNA does not treat ions explicitly. Instead, any effects of dissociation are implicitly embedded in the binary training data. The extension to explicit modeling of the ionic species is beyond the scope of this work. Also, data on polymers were not considered, so HANNA should not be used for polymer systems (note that this is currently also not possible for technical reasons, as ChemBERTa-2 does not support the correct conversion of SMILES into embeddings for molecules consisting of more than 512 tokens[42,43]). Furthermore, HANNA was not trained on solutions of strong electrolytes (except ionic liquids), so HANNA should not be used for these systems.

In the extensive tests we have conducted, HANNA consistently produced activity coefficients that yielded plausible phase behavior. The only exception to this was a small fraction (<1%) of the binary LLE systems, for which we found that HANNA predicts the existence of two separate LLE phase regions at elevated temperatures. Although this is thermodynamically possible, it has, to our knowledge, not been experimentally observed in these systems. Most of the affected systems contain methanol, ethanol, or water. We speculate that the ChemBERTa-2[43] tokenization for methanol (SMILES: "CO") and ethanol (SMILES: "CCO") into "C" (equivalent to the SMILES for methane) and "O" (equivalent to the SMILES for water) results in embeddings that cannot fully capture the properties of these small molecules. In the future, this could be overcome by introducing a distinct token for methanol and/or water; a similar strategy was used in mod. UNIFAC, which has dedicated groups for methanol and water. Despite the fundamentally different architecture and training data, we observe a similar behavior for predictions from GE-GNN, where some systems containing methanol or water are also affected. Note that Antolović et al.[44] have recently encountered similar anomalies for predictions of LLE with COSMO-SAC.

While the Lipschitz regularization promotes the smoothness of the HANNA predictions with respect to a variation of the inputs (components, temperature, and composition), excessively strong regularization overly restricts the model's flexibility and reduces its predictive performance. With the chosen regularization, HANNA, in some cases, exhibits small fluctuations of the predicted activity coefficients with respect to the composition. Because this only affects nearly ideal mixtures, i.e., with very small activity coefficients, it does not compromise HANNA's practical applicability.

## Discussion

This work presents HANNA, an ML-based $g^E$ model that enables the thermodynamically consistent prediction of activity coefficients in liquid mixtures with an arbitrary number of components solely from the chemical formula of the constituent components. HANNA was trained on a large, diverse dataset for vapor–liquid equilibria (VLE), activity coefficients at infinite dilution (ACI), excess enthalpies (HE), and liquid–liquid equilibria (LLE), resulting in a model with broad applicability.

The results of HANNA were systematically compared to those from state-of-the-art models from the UNIFAC family and other ML-based models from the literature for binary and ternary mixtures, overall demonstrating a better performance of HANNA - despite an important bias of the comparison in favor of the benchmark models: while all results for HANNA were true predictions (none of the systems in our test set was used for training HANNA), this is not true for the other models, for which we must assume that many of the systems in the test set were also used for the training. Even more importantly, HANNA has much broader applicability than UNIFAC, as only the

component SMILES are required, which are always available. Similarly, HANNA is much more broadly applicable than all other ML-based models in the literature, which are, among others, restricted to a single temperature.

HANNA is the first $g^E$ model in which the entire relevant thermodynamic knowledge is embedded in a neural network as hard constraints. It therefore fully leverages the flexibility of neural networks, but applies it only within the bounds permitted by physical principles. This flexibility enables HANNA to simultaneously describe VLE and LLE with high accuracy with a single parameterization.

A ready-to-use version of HANNA, trained on all available data for binary mixtures (including the data retained as test data for the comparison in the present work), is available on GitHub. Furthermore, we also provide an interactive interface for HANNA on our website MLPROP. We encourage users to give us feedback.

Several enabling technologies used within the work on HANNA merit to be highlighted, as they are useful far beyond the work described here: (1) Based on the CEM, we have developed a new surrogate solver for determining LLE phase compositions that can be incorporated in the workflow of ML model training. (2) Using the CEM to determine LLE, we have obtained phase-split predictions in a numerically robust and initialization-independent manner for thousands of systems in an automated workflow. Unlike conventional solvers, the CEM does not require careful initialization or phase-stability checks to identify the correct number of coexisting phases. (3) By applying a geometric projection method, a model for predicting properties of binary mixtures can be seamlessly extended to multicomponent mixtures without the need for additional parameters.

HANNA represents a major advancement in predicting the thermodynamic properties of liquid mixtures. Because we release both the full model architecture and trained parameters as open source, process engineers can integrate HANNA directly into existing workflows and simulation software. Users who prefer to continue to use conventional thermodynamic models can, e.g., fit NRTL or UNIQUAC parameters to HANNA's predictions. These fitted parameters can directly be used in established simulation frameworks without additional implementation effort, albeit typically with some loss in accuracy relative to the original HANNA predictions. A routine for fitting NRTL parameters to HANNA is also provided in MLPROP. Additionally, HANNA is interesting for material design, e.g., for identifying promising solvents for chemical reactions and separation processes like extraction, extractive distillation, absorption, and crystallization.

The current version of HANNA is already a widely applicable and accurate $g^E$ model that outperforms current state-of-the-art benchmark models. To further extend the scope of HANNA, we plan to address the following aspects.

So far, we have used data from measurements of VLE, ACI, LLE, and HE for training HANNA. In future work, data on solid–liquid equilibria (SLE) could also be included. These data would extend the training data's temperature range at its lower end[45]. Moreover, only binary data have been used for training HANNA, and the $g^E$ for multicomponent mixtures is found by a geometric projection of the $g^E$ of all constituent binary subsystems. In future work, new projection strategies could be developed for this purpose, and it could be considered to include ternary data in the fitting process. This has a bad reputation for parameterizing physics-based $g^E$ models, but it could be worth testing the option for hybrid models like HANNA. Even the introduction of ternary parameters could be considered.

HANNA currently uses the pre-trained transformer model ChemBERTa-2[43] to obtain the molecular embeddings from SMILES strings. ChemBERTa-2 was trained on a wide range of basic chemical data[43] and could be fine-tuned for the present application; however, this would significantly increase the computational cost of training without guaranteeing better results. As described above, we have observed minor deficiencies that are likely associated with issues of

ChemBERTa-2 and may be resolved by a workaround within Chem-BERTa-2, without the need for a complete revision. Specifically, one of the unused tokens in ChemBERTa-2 could be assigned to the water molecule, such that water and, e.g., hydroxyl groups no longer share the same token. Another option would be to provide different molecular embeddings, e.g., based on the SELFormer[46] that uses SELFIES[47] instead of SMILES, or even use learnable embeddings from a GNN[48].

Last but not least, we look forward to user feedback, which we will gladly use to further improve HANNA in ways we may not yet have anticipated.

## Methods

### HANNA architecture

The algorithm in Box 1 explains the forward pass of the HANNA architecture, starting with the temperature, the composition, and the molecular embeddings of an arbitrary $N$-component mixture of interest as input and providing predictions for the activity coefficients of the $N$ components and the scaled molar excess Gibbs energy in the mixture as output.

The input of HANNA contains the numerical molecular embeddings $\mathbf{e}_i$ of the $N$ components that make up the mixture to be modeled, summarized in $\mathbf{E} = [\mathbf{e}_1, ..., \mathbf{e}_N]$, which are calculated individually with the transformer-based model ChemBERTa-2 from the respective component's SMILES (canonized with RDKit[49]). We use the 77M-MTR variant of ChemBERTa-2 from Hugging Face[42] with a slightly corrected tokenizer[32]. Additionally, HANNA requires the temperature $T$ and the mole fractions $x_1, ..., x_{N-1}$ of $N-1$ components as input. The HANNA architecture comprises three feed-forward neural networks (FFNNs), namely, the EmbeddingNetwork, the MixtureNetwork, and the PropertyNetwork. The EmbeddingNetwork consists of one layer, while the other two networks consist of two layers. In all cases, we use the Sigmoid Linear Unit activation function.

All layers are modified linear layers with 96 nodes (except for the input layer of the MixtureNetwork, which consists of 98 nodes, as the

---

## BOX 1

# Forward pass of HANNA

**Input:** temperature $T$, mole fractions $\mathbf{x} = [x_1, ..., x_{N-1}]$, and ChemBERTa-2 embeddings $\mathbf{E} = [\mathbf{e}_1, ..., \mathbf{e}_N]$.

1. The inputs are scaled according to

$$T^* = \text{TemperatureScaler}(T)$$
$$\mathbf{e}_i^* = \text{EmbeddingScaler}(\mathbf{e}_i)$$

2. The mole fraction of the $N$th component is obtained from the summation condition:

$$x_N = 1 - \sum_{i=1}^{N-1} x_i$$

3. The molecular embeddings are refined as

$$\boldsymbol{\theta}_i = \text{EmbeddingNetwork}(\mathbf{e}_i^*)$$

4. The similarity score of all component pairs $i$–$j$ is computed as

$$R_{ij} = \exp\left(-\beta \left\| \boldsymbol{\theta}_i - \boldsymbol{\theta}_j \right\|^2\right) \text{ with } \beta = 100$$

5. Identical components are lumped according to

$$\tilde{x}_i = \sum_{j=1}^{N} x_j R_{ij}$$

6. For all binary subsystems with $1 \leq i < j \leq N$, the following steps are performed:

   (a) The Muggianu projection is computed as

   $$X_i^{(ij)} = \frac{1 + \tilde{x}_i - \tilde{x}_j}{2}, \ X_j^{(ij)} = \frac{1 + \tilde{x}_j - \tilde{x}_i}{2}$$

   (b) Subsystem feature vectors are constructed as

   $$\mathbf{c}_i^{(ij)} = [\boldsymbol{\theta}_i, X_i^{(ij)}, T^*]$$
   $$\mathbf{c}_j^{(ij)} = [\boldsymbol{\theta}_j, X_j^{(ij)}, T^*]$$

   (c) Within each binary subsystem, the mixture network is evaluated according to

   $$\boldsymbol{\alpha}_i^{(ij)} = \text{MixtureNetwork}(\mathbf{c}_i^{(ij)})$$
   $$\boldsymbol{\alpha}_j^{(ij)} = \text{MixtureNetwork}(\mathbf{c}_j^{(ij)})$$

   (d) The binary interaction is computed as

   $$\phi_{ij} = \text{PropertyNetwork}\left(\boldsymbol{\alpha}_i^{(ij)} + \boldsymbol{\alpha}_j^{(ij)}\right)$$

   (e) Consistency for pure components is enforced via

   $$q_{ij} = \phi_{ij}(1 - R_{ij})$$

7. The scaled molar excess Gibbs energy is then obtained as

$$\frac{g^{\text{E}}}{\text{R}T} = \sum_{i=1}^{N-1} \sum_{j=i+1}^{N} x_i x_j q_{ij}$$

8. The activity coefficients for $i = 1, ..., N-1$ are computed as

$$\ln \gamma_i = \frac{g^{\text{E}}}{\text{R}T} + \frac{\partial}{\partial x_i}\left(\frac{g^{\text{E}}}{\text{R}T}\right) - \sum_{j=1}^{N-1} x_j \frac{\partial}{\partial x_j}\left(\frac{g^{\text{E}}}{\text{R}T}\right)$$

9. The activity coefficient of the final component is computed as

$$\ln \gamma_N = \frac{g^{\text{E}}}{\text{R}T} - \sum_{j=1}^{N-1} x_j \frac{\partial}{\partial x_j}\left(\frac{g^{\text{E}}}{\text{R}T}\right)$$

**Output:** activity coefficients $\ln \gamma_1, ..., \ln \gamma_N$, scaled molar excess Gibbs energy $g^{\text{E}}/(\text{R}T)$.

---

temperature and the projected mole fraction of the respective component are concatenated to the output of the EmbeddingNetwork) to which a Lipschitz regularization (cf. section "Training of HANNA") is applied. HANNA computes the molar excess Gibbs energy for the $N$-component mixture of interest through a geometric projection from the binary subsystems[40,41,50–52]. For this purpose, the mole fractions of the $N$-component mixture are projected onto all $N(N-1)/2$ binary subsystems (cf. loop in step 6 of the algorithm in Box 1). In this work, we use the Muggianu projection[33], which, unlike other projection methods, does not exhibit singular points at infinite dilution[53,54]. The projected mole fractions $X_i^{(ij)}$ and $X_j^{(ij)}$ in the binary subsystem $i$–$j$ are given by:

$$X_i^{(ij)} = \frac{1 + \tilde{x}_i - \tilde{x}_j}{2} \text{ and } X_j^{(ij)} = \frac{1 + \tilde{x}_j - \tilde{x}_i}{2} \quad (1)$$

whereby $\tilde{x}_i$ and $\tilde{x}_j$ are the corrected mole fractions of component $i$ and $j$, respectively (cf. below). The projection strictly fulfills the summation condition in each binary subsystem, i.e., $X_i^{(ij)} + X_j^{(ij)} = 1$. Furthermore, if a binary mixture is studied, the projected mole fractions reduce to the corrected mole fractions, i.e., $X_i^{(ij)} = \tilde{x}_i$ and $X_j^{(ij)} = \tilde{x}_j$, ensuring a consistent transition of the HANNA predictions going from a binary to a multi-component mixture. In contrast to the original publication[33], we correct the original mole fractions $x_i$ in the $N$-component mixture before the projection as follows (cf. steps 4 and 5 of the algorithm in Box 1):

$$\tilde{x}_i = \sum_{j=1}^{N} x_j R_{ij} \text{ with } R_{ij} \in [0,1] \quad (2)$$

With this correction, we lump identical components, indicated by a component similarity score $R_{ij} = 1$, into a single component, thereby ensuring that, e.g., for a system of three components A, B, and C, where B is identical to C, the exact same results are obtained as for the binary system of A and B (when accounting for the lumped mole fractions). To compute the component similarity score $R_{ij}$, we apply the radial basis function (RBF) on pairs of the refined molecular embeddings. By setting the hyperparameter $\beta$ of the RBF to the high value of 100, we ensure that only extremely similar components result in a similarity score of $R_{ij} = 1$, so only components that we consider truly identical are lumped together.

Finally, the molar excess Gibbs energy $g^E$ of the $N$-component mixture is calculated as follows[41]:

$$\frac{g^E}{RT} = \frac{1}{RT} \sum_{i=1}^{N-1} \sum_{j=i+1}^{N} \frac{x_i x_j}{X_i^{(ij)} X_j^{(ij)}} g_{ij}^E \left( X_i^{(ij)}, X_j^{(ij)} \right)$$
$$= \sum_{i=1}^{N-1} \sum_{j=i+1}^{N} x_i x_j q_{ij} \left( X_i^{(ij)}, X_j^{(ij)} \right) \quad (3)$$

where $g_{ij}^E$ is the molar excess Gibbs energy of the binary subsystem $i$–$j$ and $q_{ij}$ is the learned binary interaction between component $i$ and $j$ in the respective binary subsystem. $q_{ij}$ is calculated using a deep set architecture[55] including a summation aggregation (cf. step 6d of the algorithm in Box 1) to ensure the permutation invariance of the predicted molar excess Gibbs energy, i.e., to ensure that the order of the components in the input of HANNA does not influence its results. For technical reasons, namely, to prevent divisions by zero in Eq. (3) if one of the projected mole fractions approaches zero, we directly predict the binary interaction term $q_{ij}$ with HANNA for each binary subsystem in the projected space (cf. step 6e of the algorithm in Box 1). Here, again, the component similarity score $R_{ij}$ is used, ensuring thermodynamically sound predictions by enforcing $q_{ij} = 0$ and, ultimately, $g_{ij}^E = 0$ for identical components $i$ and $j$. Using the Autograd functionality of PyTorch[56], the activity coefficients for all $N$ components are

**Table 2 | Overview of the final datasets used in this work**

| Dataset | | Data points | Systems | Components |
|---|---|---|---|---|
| Binary | TPXY | 193,484 | 5,496 | 985 |
| | TPX | 174,409 | 4,885 | 1,005 |
| | ACI | 124,058 | 30,428 | 2,229 |
| | LLE | 76,041 | 5,787 | 2,330 |
| | HE | 256,489 | 9,574 | 1,347 |
| Ternary | TPXY | 43,021 | 1,194 | 374 |
| | TPX | 20,326 | 448 | 252 |
| | ACI | 5,832 | 520 | 133 |
| | LLE | 89,961 | 6,407 | 1,413 |
| | HE | 31,455 | 662 | 211 |
| Quaternary | VLE | 3,763 | 132 | 87 |
| | LLE | 19,255 | 1,036 | 340 |

All data were taken from the DDB[34].

finally derived from $g^E$ of the $N$-component mixture (cf. steps 8 and 9 of the algorithm in Box 1), ensuring strict Gibbs-Duhem consistency[57].

## Data

The experimental data used to train and evaluate HANNA in this work were taken from the 2025 version of the DDB[34]. Data points labeled as poor quality by the DDB's internal criteria were excluded. Furthermore, only components for which a canonical SMILES string could be generated with RDKit[49] using mol-files from the DDB were considered. Table 2 gives an overview of the final datasets of the different data types for binary, ternary, and quaternary systems. A detailed description of each data type is given below.

Two types of experimental VLE data were used in this work: On the one hand, we considered data points in the DDB[34] that contain all information on temperature, pressure, as well as liquid and vapor phase compositions. These are labeled as TPXY data. From the TPXY data, activity coefficients for all components $i$ were calculated with the extended Raoult's law:

$$\gamma_i = \frac{p y_i}{p_i^s x_i} \quad (4)$$

Here, $\gamma_i$ is the activity coefficient of component $i$ normalized according to Raoult's law in the mixture, $x_i$ and $y_i$ are the mole fractions of component $i$ in the liquid and vapor phase in VLE, respectively, $p$ denotes the total pressure, and $p_i^s$ is the pure-component vapor pressure of $i$ at the same temperature as the VLE data. The pure-component vapor pressures were computed using the Antoine equation with coefficients from the DDB[34]. On the other hand, we used VLE data that does not include information on the composition of the vapor phase. Consequently, these are labeled as TPX data. Because activity coefficients cannot be calculated for the TPX data, we use total pressure as the training objective for those data points. By using extended Raoult's law for all VLE data points in our work, we assume that the vapor phase can be treated as an ideal gas mixture and that the pressure dependence of the chemical potential in the liquid phase can be neglected. To justify these assumptions, all TPXY and TPX data points with pressures higher than 10 bar were excluded.

Activity coefficients at infinite dilution $\gamma_i^\infty$ normalized according to Raoult's law, called "ACI" throughout this work, were considered for binary and ternary mixtures. For binary mixtures, i.e., with pure solvents, these data could be used directly. For the activity coefficients at infinite dilution in mixed solvents, we observed several errors in the DDB, mostly interchanged solvent labels, so that these data had to be curated prior to use. Our detailed data curation protocol is described

in the following: For each ternary ACI system reported at temperature $T_{ACI}$, we compared the reported logarithmic activity coefficients at infinite dilution $\ln\gamma_3^\infty$ at the pure-solvent endpoints (i.e., $x_1 = 0$ or $x_2 = 0$) with those of the same solute in the corresponding pure solvents from the binary ACI data at temperatures within $\pm 5\,K$ of $T_{ACI}$. Systems for which no such binary reference data were available, or for which the absolute deviation exceeded a value of 0.1, were excluded. Only the remaining mixed-solvent data points ($x_1 \neq 0$ and $x_2 \neq 0$) of the retained systems were included in the final dataset.

The LLE data in the DDB contains information on the temperature $T$ and the mole fractions in the coexisting phases $x_i'$ and $x_i''$ (and $x_i'''$ in the case of a three-phase equilibrium). Here, data for binary, ternary, and quaternary mixtures were considered. For binary data, we followed the notation of the DDB and defined the phase compositions such that $x_1'$ is always the phase composition with the lower value, i.e., $x_1'' > x_1'$ always holds, which is important for training the surrogate solver, cf. section "Surrogate LLE solver". Only for roughly a quarter of the LLE data points in the DDB all phase compositions are reported at the specified temperature. For the remaining data points, only the composition of one phase is available. Nonetheless, we included these data points in our dataset. Because HANNA does not consider the pressure dependence of the excess Gibbs energy and thus the activity coefficients, we excluded all LLE data points with pressures above 50 bar.

Finally, we also extracted experimental data for the excess enthalpies $h^E$ of binary and ternary mixtures from the DDB. Each data point consists of the temperature $T$, the mole fraction $x_1$, and the excess enthalpy $h^E$.

## Data splitting and scaling

Following a cross-validation approach, we randomly split the datasets for binary mixtures system-wise into 10 folds, each representing a mutually exclusive test set of 10% of the binary systems. A stratified split strategy was employed, such that the share of each data type is roughly equal for all folds. The remaining data (90% of the systems) of each fold were again split system-wise into a training set (90%) and a validation set (10%). In the data splits, all types of data for a particular system were put exclusively within a single set; for instance, if LLE data for a particular system were in the test set, all TPXY, TPX, ACI, and HE data for this system were also in the test set. This strategy enabled the use of all available data as unseen test data for the evaluation of HANNA.

Furthermore, an additional data split was carried out, where 95% of the binary systems were used in the training set and 5% in the validation set (i.e., no test set was withheld). This split, denoted as "full data split" in the following, was used to train the model that was subsequently evaluated on the data for ternary and quaternary systems and is also deployed as the "final model" with this work.

The molecular embeddings from ChemBERTa-2, as well as the temperature, were scaled using the scikit-learn[58] StandardScaler(). Individual scalers were fitted to each training dataset to prevent any information leakage.

## Surrogate LLE solver

From the binary TPXY data points, activity coefficients can be calculated directly using the extended Raoult's law, cf. Eq. (4). Hence, the training of HANNA on these data is straightforward using a loss function for the predicted activity coefficients, cf. section "Training of HANNA". In contrast, it is not possible to explicitly calculate activity coefficients from the binary experimental LLE data. Using HANNA, one could obtain phase compositions for a given system and state point using iterative solvers[35] and compare them to experimental data. However, integrating iterative solvers into a deep learning framework is tedious because gradients are not easily trackable and, moreover, it would significantly slow down the training process. To overcome this, we developed a differentiable surrogate solver to estimate the binary

LLE phase compositions resulting from the activity coefficients predicted by HANNA. This surrogate solver is realized by an FFNN that mimics the CEM[37–39] and maps a discretized graph of the scaled molar Gibbs energy of mixing

$$\frac{\Delta g_{mix}}{RT} = x_1 \ln(x_1\gamma_1) + x_2 \ln(x_2\gamma_2) \tag{5}$$

onto the compositions of the two coexisting liquid phases $x_1'$ and $x_1''$. During the training of HANNA, the surrogate solver allows the model to directly learn from experimental LLE data while maintaining full compatibility with gradient-based optimization. The architecture and training process of the surrogate solver are described in the following.

As input, the surrogate solver gets a vector of $\Delta g_{mix}/(RT)$ values calculated with HANNA at 101 uniformly spaced discrete compositions from $x_1 = 0$ to $x_1 = 1\,mol\,mol^{-1}$ at constant temperature $T$. The $\Delta g_{mix}/(RT)$ values from HANNA are then processed through three fully connected layers with a hidden size of 64, each followed by the Rectified Linear Unit (ReLU) activation function. The output layer yields two values that correspond to $x_1'$ and $x_1''$ (we always define $x_1'$ as the phase composition with the lower value, cf. section "Data"), which are subsequently processed through a sigmoid function to constrain the mole fractions between 0 and 1. To align with the demand of thermodynamic consistency of the model, the surrogate solver must be permutation-equivariant with respect to the order of the molecules in the input, i.e., if the discretized Gibbs energy of mixing is given in reverse order, it must yield $1 - x_1''$ and $1 - x_1'$. To enforce this, upon calling the forward function of the surrogate solver with a discretized $\Delta g_{mix}/(RT)$ vector, the phase compositions are calculated twice, both from the original vector and the reversed vector. The two values obtained for $x_1'$ and $x_1''$ are then averaged. Besides guaranteeing permutation-equivariance, this procedure also prevents any bias in the order of the training data from negatively impacting the training.

The surrogate solver was trained only on synthetic data, namely, predictions of $\Delta g_{mix}/(RT)$ obtained with mod. UNIFAC. More specifically, for every experimental LLE data point in our dataset within the mod. UNIFAC horizon, we calculated $\Delta g_{mix}/(RT)$ for the 101 discrete uniformly spaced compositions at the temperature of the respective data point with mod. UNIFAC. We then used the CEM to determine the corresponding phase compositions, which served as the training objective of our surrogate solver. If no phase split was found, the data point was dropped. We used the same system-wise split as for the experimental data (cf. section "Data splitting and scaling") to split the synthetic surrogate solver data into training, validation, and test sets. This prevents information leakage through the surrogate solver when training HANNA. We trained separate surrogate solvers for each fold on the synthetic $\Delta g_{mix}/(RT)$ values of the corresponding LLE training data. The surrogate solvers were trained for 200 epochs, using the Adam optimizer with the MSE loss and the OneCycleLR scheduler (max_lr = 0.01), and the best model was selected based on the validation loss (the validation dataset was generated in the same manner as the training set). Results of the surrogate solver training are provided in the Supplementary Note 1, along with an example of its functionality given in Supplementary Fig. 1.

## Training of HANNA

HANNA was trained by minimizing the total loss $\mathcal{L}_{total}$, defined as:

$$\mathcal{L}_{total} = \frac{1}{N_b}\Big(\mathcal{L}_{TPXY} + w_{TPX}\,\mathcal{L}_{TPX} + \mathcal{L}_{ACI} + w_{LLE}\,\mathcal{L}_{LLE} + w_{HE}\,\mathcal{L}_{HE} + w_{Gibbs}\,\mathcal{L}_{Gibbs} + w_{Lips}\,\mathcal{L}_{Lips}\Big) \tag{6}$$

where $N_b$ is the batch size, $\mathcal{L}_{TPXY}$, $\mathcal{L}_{TPX}$, $\mathcal{L}_{ACI}$, $\mathcal{L}_{LLE}$, $\mathcal{L}_{HE}$ are the individual data losses, $\mathcal{L}_{Gibbs}$ is the Gibbs loss, $\mathcal{L}_{Lips}$ is the Lipschitz regularization,

and $w_{TPX}$, $w_{LLE}$, $w_{HE}$, $w_{Gibbs}$, $w_{Lips}$ are the corresponding weight factors that were determined in a hyperparameter grid search, cf. Supplementary Note 2 for details.

In the individual data losses, we use the SmoothL1Loss() loss function from PyTorch[56], which is less sensitive to outliers than the mean squared error loss. The parameter $\beta$ in the SmoothL1Loss() function, which controls the transition between the mean absolute and mean squared error, was empirically set for each data type individually.

$\mathcal{L}_{TPXY}$ penalizes the average deviation of the two logarithmic activity coefficients predicted with HANNA from the respective experimental values derived from the TPXY data for the binary systems from the training set:

$$\mathcal{L}_{TPXY} = \sum_{k}^{N_{TPXY}} SmoothL1Loss\left([\ln \gamma_{1,pred}, \ln \gamma_{2,pred}], [\ln \gamma_{1,exp}, \ln \gamma_{2,exp}]\right) \text{ with } \beta = 1.0 \tag{7}$$

where $N_{TPXY}$ is the number of TPXY data points in the training batch. Similarly, $\mathcal{L}_{ACI}$ takes into account errors in the prediction of the activity coefficients at infinite dilution $\ln \gamma_i^\infty$ for the binary systems from the training set:

$$\mathcal{L}_{ACI} = \frac{1}{2} \sum_{k}^{N_{ACI}} SmoothL1Loss\left(\ln \gamma_{i,pred}^\infty, \ln \gamma_{i,exp}^\infty\right) \text{ with } \beta = 2.0 \tag{8}$$

where $N_{ACI}$ is the number of ACI data points in the batch. The factor $\frac{1}{2}$ takes into account that in $\mathcal{L}_{ACI}$, only the activity coefficient at infinite dilution contributes to the loss calculation, whereas in $\mathcal{L}_{TPXY}$, the errors of both activity coefficients of the binary mixture are averaged before contributing to the loss. $\mathcal{L}_{TPX}$ accounts for the difference of the predicted logarithmic total pressures to the experimental ones on the $N_{TPX}$ data points:

$$\mathcal{L}_{TPX} = \sum_{k}^{N_{TPX}} SmoothL1Loss\left(\ln(p_{pred}/bar), \ln(p_{exp}/bar)\right) \text{ with } \beta = 1.0 \tag{9}$$

The predicted logarithmic total pressures are calculated from the predicted activity coefficients and the pure component vapor pressures:

$$\ln(p_{pred}/bar) = \ln\left(\frac{p_1^s x_1 \gamma_{1,pred} + p_2^s x_2 \gamma_{2,pred}}{bar}\right) \tag{10}$$

$\mathcal{L}_{LLE}$ denotes the loss between the LLE phase compositions in mole fractions predicted by HANNA and the experimental values in the batch obtained through

$$\mathcal{L}_{LLE} = \sum_{k}^{N_{LLE}} m_k \cdot SmoothL1Loss\left([x'_{1,pred}, x''_{1,pred}], [x'_{1,exp}, x''_{1,exp}]\right) \text{ with } \beta = 0.35 \tag{11}$$

where $N_{LLE}$ is the number of LLE data points in the batch and $m_k \in \{0, 1\}$ is a binary mask that indicates if the respective data point is considered or not (cf. below for details). To obtain the predicted phase compositions $x'_{1,pred}, x''_{1,pred}$ for a data point during training, the pre-trained surrogate solver (cf. section "Surrogate LLE solver") is provided with the discretized molar Gibbs energy of mixing curve for the respective system and temperature. For the LLE data points with only one available phase composition, the loss term in Eq. (11) is calculated only on this phase composition. Since the surrogate solver is only trained on positive samples (i.e., binary LLE systems at state points with a

miscibility gap) and cannot identify systems without miscibility gap, it will also falsely predict a phase split for a $\Delta g_{mix}$ that does not fulfill the necessary condition for a phase split, which states

$$\left(\frac{\partial^2 \Delta g_{mix}}{\partial x_1^2}\right)_{T,p} < 0 \tag{12}$$

i.e., the second derivative of $\Delta g_{mix}$ is negative somewhere in the composition space. This is unphysical and could potentially negatively impact the training process. Therefore, during the training, we account for the LLE loss of binary systems only if the necessary condition in Eq. (12) is true for at least one discretized composition; this is reflected in Eq. (11) through the binary mask $m_k$, where $m_k = 1$ for data points that exhibit an LLE and $m_k = 0$ for those that do not. The required second derivatives of $\Delta g_{mix}$ were calculated by automatic differentiation. $\mathcal{L}_{HE}$ describes the loss on the $N_{HE}$ excess enthalpy data points:

$$\mathcal{L}_{HE} = \sum_{k}^{N_{HE}} SmoothL1Loss\left(h_{pred}^E, h_{exp}^E\right) \text{ with } \beta = 2.0 \tag{13}$$

The predicted excess enthalpy is calculated using[35]

$$h_{pred}^E = -RT^2\left(\frac{\partial}{\partial T}\frac{g_{pred}^E}{RT}\right)_x = -\frac{RT^2}{s_T}\left(\frac{\partial}{\partial T^*}\frac{g_{pred}^E}{RT}\right)_x \tag{14}$$

The factor $1/s_T$ in Eq. (14) follows from the chain rule because HANNA uses the scaled temperature $T^* = (T - \mu_T)/s_T$ as input.

The $\mathcal{L}_{LLE}$ described above alone does not penalize false negative LLE predictions. If HANNA fails to predict a liquid−liquid phase split, the necessary condition in Eq. (12) is not fulfilled, and the binary mask is $m_k = 0$, cf. Eq. (11). To provide a learning incentive towards a correct detection of these phase splits, the additional loss term $\mathcal{L}_{Gibbs}$ was introduced:

$$\mathcal{L}_{Gibbs} = \sum_{k}^{N_{LLE}} \max\left(0, \min_d S_k(T_k, x_{1,d})\right) \tag{15}$$

Here,

$$S_k(T_k, x_{1,d}) = \frac{1}{RT_k}\left(\frac{\partial^2 \Delta g_{mix}(T_k, x_{1,d})}{\partial x_1^2}\right)_T \tag{16}$$

is the second derivative of the molar Gibbs energy of mixing for data point $k$ from the LLE training set evaluated at the composition $d$ on the discretized grid of $\Delta g_{mix}$. Eq. (15) identifies the minimum of the second derivative of $\Delta g_{mix}$ within the discretized composition space for each LLE training data point. If this minimum is negative, a miscibility gap is predicted by the model (cf. Eq. (12)) and the Gibbs loss is zero. Note that in this case ($m_k = 1$), the deviation to experimental data is accounted for by $\mathcal{L}_{LLE}$. If the minimum is positive, the miscibility gap is not identified by HANNA, and, consequently, the model is penalized by adding the respective value of $S_k$ to the total loss.

Besides the data losses and the Gibbs loss, $\mathcal{L}_{Lips}$ is included in the total loss function as a Lipschitz regularization term, controlling the Lipschitz constant of HANNA. Generally, the Lipschitz constant ($c \geq 0$) in the 2-norm is defined as

$$\| f(x_1) - f(x_0) \|_2 \leq c \, \| x_1 - x_0 \|_2 \tag{17}$$

and provides an upper bound for the change of a function from $f(x_0)$ to $f(x_1)$ by changing the function input from $x_0$ to $x_1$; it is, therefore, a proxy for the smoothness of a function. For fully connected neural

networks with $L$ layers, a loose upper bound

$$c = \prod_{l=1}^{L} c_l = \prod_{l=1}^{L} \| \mathbf{W}_l \|_2 = \prod_{l=1}^{L} \sigma_{\max}(\mathbf{W}_l) \tag{18}$$

is obtained by the product of the spectral norms $\| \mathbf{W}_l \|_2$ of the weight matrices of each layer $l$, if the Lipschitz constants of the activation functions are neglected[59]. For HANNA, the influence of the summation of the two mixture embeddings (cf. step 6d of the algorithm in Box 1) on the total Lipschitz constant is also neglected, since it does not contain any learnable parameters. The spectral 2-norm $\| \mathbf{W}_l \|_2$, in turn, equals the maximum singular value $\sigma_{\max}(\mathbf{W}_l)$ of the weight matrix $\mathbf{W}_l$[60].

Following the approach from ref. 59, all networks of HANNA are realized as modified linear layers with learnable Lipschitz constants $c_l^*$. In these layers, the raw weight matrices $\mathbf{W}_{\text{raw}}$ are first normalized using the maximum singular value $\sigma_{\max}(\mathbf{W}_{l,\text{raw}})$ (such that their Lipschitz constant equals one) and then scaled using the learnable Lipschitz constants:

$$\mathbf{W}_{l,\text{scaled}} = \frac{\mathbf{W}_{l,\text{raw}}}{\sigma_{\max}(\mathbf{W}_{l,\text{raw}})} \cdot \text{softplus}(c_l^*) \tag{19}$$

These scaled weight matrices are then used in the forward function of the modified layers as a drop-in replacement of the raw weight matrices. For computational efficiency, we approximate $\sigma_{\max}(\mathbf{W}_{l,\text{raw}})$ with a power iteration method using two iterations[60]. The Lipschitz regularization term used in the total loss function, cf. Eq. (6), is calculated via[59]

$$\mathcal{L}_{\text{Lips}} = \prod_{l=1}^{L} \text{softplus}(c_l^*) \tag{20}$$

by multiplying the total Lipschitz constants of the networks (given by the product of the layer-wise learned constants of all $L$ layers in the model and ignoring other factors, cf. above). The softplus transformation of $c_l^*$ avoids infeasible negative Lipschitz constants[59]. The hyperparameter $w_{\text{Lips}}$, cf. Eq. (6), can be used to control the learnable Lipschitz constants of the network, i.e., higher values of $w_{\text{Lips}}$ lead to lower Lipschitz constants and vice versa. For the model variants in the grid search with $w_{\text{Lips}} = 0$, we employed an L2-regularization instead to prevent exploding parameters.

The HANNA models for this work were trained for 200 epochs using a batch size of 512, the AdamW optimizer, and the OneCycleLR scheduler (max_lr = 0.01) from PyTorch. The best epoch was selected based on the weighted sum of the data losses ($\mathcal{L}_{\text{TPXY}}$, $\mathcal{L}_{\text{TPX}}$, $\mathcal{L}_{\text{ACI}}$, $\mathcal{L}_{\text{LLE}}$, and $\mathcal{L}_{\text{HE}}$) on the respective validation set, whereby only models that were trained for at least 50 epochs were considered.

After determining the optimal hyperparameters through a grid search (cf. Supplementary Note 2), we trained an ensemble of 10 models (starting from different weight initializations) for each data fold (cf. section "Data splitting and scaling"). Additionally, we trained such an ensemble for the full data split; this model is also published as the final model along with this work. Training was carried out on NVIDIA A30 GPUs; a typical training run of a single HANNA model (200 epochs) required approximately 20 h.

### Evaluation of HANNA
For the systematic evaluation of HANNA and the benchmark models, absolute deviations $\epsilon$ between the predicted and the experimental values for each data point in the considered test set were calculated.

For each TPXY data point, a mean absolute deviation of the logarithmic activity coefficients of all components $i$ in the test mixture

at the respective temperature $T$ and composition $\mathbf{x}$ was calculated:

$$\epsilon_{\text{TPXY}} = \frac{1}{N} \sum_{i=1}^{N} \left| \ln \gamma_i^{\text{pred}} - \ln \gamma_i^{\text{exp}} \right| \tag{21}$$

Here, $N$ is the number of components in the mixture ($N = 2$ for binary, $N = 3$ for ternary, $N = 4$ for quaternary mixtures).

To evaluate the model performance on the TPX data points, the absolute deviations in the logarithmic total pressures were calculated:

$$\epsilon_{\text{TPX}} = \left| \ln\left( p^{\text{pred}} / \text{bar} \right) - \ln\left( p^{\text{exp}} / \text{bar} \right) \right| \tag{22}$$

For each ACI data point, i.e., activity coefficients at infinite dilution in pure or in mixed solvents, only the logarithmic activity coefficient of the infinitely diluted solute $i$ in the solvent(s) was considered for error calculation:

$$\epsilon_{\text{ACI}} = \left| \ln \gamma_i^{\text{pred},\infty} - \ln \gamma_i^{\text{exp},\infty} \right| \tag{23}$$

The LLE predictions were evaluated through the mean absolute deviation of the phase compositions in mole fractions $x_i'$ and $x_i''$ (and $x_i'''$ for three-phase equilibria) of all components $i$ in the mixture:

$$\epsilon_{\text{LLE}} = \begin{cases} \frac{1}{2N} \sum_{i=1}^{N} \left( \left| x_{i,\text{pred}}' - x_{i,\text{exp}}' \right| + \left| x_{i,\text{pred}}'' - x_{i,\text{exp}}'' \right| \right) & \text{for } \pi = 2 \\ \frac{1}{3N} \sum_{i=1}^{N} \left( \left| x_{i,\text{pred}}' - x_{i,\text{exp}}' \right| + \left| x_{i,\text{pred}}'' - x_{i,\text{exp}}'' \right| + \left| x_{i,\text{pred}}''' - x_{i,\text{exp}}''' \right| \right) & \text{for } \pi = 3 \end{cases} \tag{24}$$

Here, $\pi$ denotes the number of liquid phases. For the HE data, the absolute deviation of the excess enthalpies in kJ mol$^{-1}$ was used as evaluation metric:

$$\epsilon_{\text{HE}} = \left| \frac{h_{\text{pred}}^{\text{E}} - h_{\text{exp}}^{\text{E}}}{\text{kJ mol}^{-1}} \right| \tag{25}$$

From the data point-wise error scores described above, system-wise mean absolute error $\text{MAE}_{\text{sys}}$ scores were calculated by averaging over all data points (at all temperatures and compositions) for each binary, ternary, or quaternary system. The system-wise error scores are favorable as they are independent of the number of available data points for a particular system and do not over-emphasize systems with many data points, such as (ethanol + water). Therefore, they are more suitable to evaluate the performance of the models to generalize over unseen systems and components.

The HANNA predictions shown in the evaluation of this work were obtained using ensembles of 10 individual models. For the binary data, we used the 10 independently trained ensembles and evaluated them on their respective test set. For the multi-component data, the final model, i.e., the ensemble trained on the full data split, was used.

## Data availability
The thermodynamic data used in this study are available under restricted access because they were obtained from the Dortmund Data Bank (DDBST GmbH, Oldenburg, Germany) under a proprietary license and therefore cannot be shared by the authors. Access to these data can be obtained directly from DDBST GmbH, subject to a license agreement and associated fees. Source data are provided with this paper.

## Code availability
The HANNA model class and trained parameters of the final model are available in a public Zenodo repository[61]. Example code to generate predictions for excess Gibbs energies, activity coefficients, and excess

enthalpies in binary and multi-component mixtures is provided in the repository.

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

## Acknowledgements

The authors gratefully acknowledge financial support by the Carl Zeiss Foundation through the project "Process Engineering 4.0" (grant number P2018-02-002, H.H., F.J.); by the German Research Foundation (DFG) through the Emmy-Noether program (grant number 528649696, F.J.), Priority Program SPP 2331 "Machine Learning in Chemical Engineering" (grant numbers 466387255, J.B.; 553345933, F.J.), Priority Program SPP 2363 "Molecular Machine Learning" (grant number 497201843, F.J.), and Research Training Group GRK 2908 "WERA" (grant number 503479768; F.J.); and by the National Science Foundation (NSF) (grant numbers IIS-2047418 and IIS-2007719, S.M.) and the NSF LEAP Center (grant number STC-2019625, S.M.). Model training and evaluation was carried out on the high-performance computer Elwetritsch at RPTU (grant RPTU-MLVT; F.J.). The authors thank Jan Rittig for providing us with the parameter sets of GDI-GNN and GE-GNN for evaluation.

## Author contributions

M.H.: conceptualization; methodology; software; validation; formal analysis; investigation; data curation; writing—original draft; visualization. T.S.: conceptualization; methodology; software; validation; formal analysis; investigation; data curation; writing—original draft. Q.G.: methodology; software. writing—review and editing. J.B.: methodology; writing—review and editing. S.M.: conceptualization; methodology; writing—review and editing. H.H.: conceptualization; writing—review and editing; supervision; project administration; funding acquisition. F.J.: conceptualization; methodology; writing—review and editing; supervision; project administration; funding acquisition.

## Funding

## Competing interests

The authors declare no competing interests.
