## [Transparent Peer Review file · Nature Communications]

Thermodynamically consistent machine learning model for excess Gibbs energy

Corresponding Author: Professor Fabian Jirasek

Version 0:

Reviewer comments:

Reviewer #1

(Remarks to the Author)

In the present manuscript a generalization of a machine learning (ML) based gE model, called HANNA, is presented. This tool could be used for the prediction of VLE and LLE of binary and multicomponent systems only from the molecular structure of the components, avoiding a model for gE or the activity coefficient.

The manuscript addresses a topic of great relevance and offers valuable insights that could guide future research in this field. The data analysis, interpretation, and conclusions are sound. In my opinion, this paper deserves to be published and will have a remarkable impact in this research area. Below just some comments:

One of the weak points of the method is the extrapolation from binary to multicomponent systems using the geometric projection of the binary gE. Also, not having used ternary and quaternary data for the training, drastically could reduce the possibility of obtaining reliable results for many multicomponent systems. For engineering calculations, the qualitative description of phase equilibrium behaviour is usually not enough (not even a rough quantitative one), as for example the island type LLE prediction presented in Figure 2c. For equipment and process design high accuracy is required. This poses a great challenge for multicomponent systems with different types of equilibrium regions involving vapor, liquid and anhydrous or hydrated solid phases (VLE, LLE, VLLE, LLSE, LLShSa,...). If "special" components are considered as polymers, ionic liquids, etc... the complexity could be even higher.

The paper does not indicate clearly the type of systems that can be addressed by HANNA. I have tested the interactive interface MLPROP, provided by the authors, to run some checks. For example, ternary type III (Treybal classification) with three liquid phases in equilibrium (LLLE tie-triangle) and island type ternary systems at constant temperature can be predicted (at least qualitatively). However, it is truth that the results in some cases could be considerably deviated from the experimental ones (source DECHEMA Chemistry Data Series). For example, I have found discrepancies in the following arbitrarily chosen systems:

- a) 1-Hexanol(1) + Nitromethane(2) + Water(3) at T=294K (type III with LLLE)
- b) acetic acid (1) + dimethyl formamide (2) + cyclohexene (3) 291.15K (island)
- c) dichloroacetic acid (1) + 1-methyl-2-pyrrolidinone (2) + 1-hexene (3) 293.15 (island)
- d) dichloroacetic acid (1) + N-formylpiperidine (2) + cyclohexane (3) 298.15K (island)

The goal of this work is very ambitious, and so one can imagine that incorrect results will be obtained for some (or even many!) predictions of multicomponent systems at this stage of the research. Along with the interface MLPROP the authors indicate (with very good judgment): "no warranty is given that the provided tool here is error-free, or that the results obtained with the tool are correct". Regardless of this, the relevant is that this study identifies a promising direction for further investigation in phase equilibria.

(Remarks on code availability)

Reviewer #2

(Remarks to the Author)

The authors present an update to their prediction model for activity coefficients (HANNA). The new model is trained on VLE and LLE data and can be evaluated for multicomponent mixtures using a simple but effective geometric projection method

that provides physically meaningful results on the binary edges of the composition space. Overall this is an important and novel contribution that tackles limitations of existing models. However, I see some shortcomings that the authors should address before publication.

Overall capability and impact of the proposed method

- The title of the manuscript mentions an expression for the excess Gibbs energy, which the manuscript never delivers. Of course, it is possible to write down an expression for any neural network in theory, but in practice that is cumbersome. I invite the authors to rethink the expectations they set with the title.

- While the authors do not spare superlatives ("excellent predictions", "clearly outperforming", "superior performance", "entire relevant physical performance") the presented results are not fully backing those claims: The MAE is slightly smaller compared to UNIFAC ("clearly outperforming") and worsens again somewhat when evaluated on systems that UNIFAC can't model ("retains its accuracy"). For ACI the predictions are way better, but UNIFAC was not developed for that and should (at least responsibly) not be used for it. The authors keep stating that UNIFAC fails on LLEs, but in their own results (based on the MAE) UNIFAC seems to work reasonably well for LLE? Of course the comparison is flawed, because many of the systems used in the evaluation were used in the determination of UNIFAC parameters, but on the other hand, UNIFAC is a much simpler model. Why not compare it to different ML models instead or as well? Wouldn't it be interesting to see how it compares to soft-constrained models or ones that are not trained on LLE data? The authors should dial back their claims or back them up with meaningful data.

- Page 5: How are the examples in Fig. 2b-d chosen (and Fig. 3b-c)? Without any indication on the decision process or at least an indication of the MAE of the systems, how are the readers not supposed to think that those are cherry-picked for the best agreement with experimental data? Please elaborate.

- Page 8: "It will improve process design and optimization, and ease their application" -> This ties in with the point above regarding the title. The model itself is fine as it is, but the authors don't describe how it would be used in design or optimization. Are you suggesting that simulation software developers will include the NN architecture of HANNA into their products? Or is it not more realistic that process engineers will fit their favorite gE model to the HANNA outputs in which case the advantage of the additional flexibility is obsolete. Please elaborate.

Literature context

- Page 2: The literature discussion at the bottom of page 2 leaves out relevant work on the prediction of VLEs with activity coefficient/gE models. What about the work on GNNs by Zavala and coauthors, soft-constrained models like the "Gibbs-Duhem-informed NN" by Rittig et al. (<https://doi.org/10.1039/D3DD00103B>), potentially Medina et al. on prediction of infinite dilution activity coefficients.

- Page 3: "The first ML-based gE-model ..." -> I suppose that might have been true at the publication of the HANNA "prototype" but by now other work has been published like <https://doi.org/10.1039/D4SC04554H>. The presented methodology reads as an extension of HANNA to include LLE predictions and multicomponent systems, which is certainly novel and exciting, but the paragraph is not clear. Please rephrase to make the reader more aware of (1) what has been published previously by the authors, (2) what other developments have been done in the meantime, (3) what is the novelty presented in this publication.

- Page 3: "enables prediction for arbitrary components" -> This is used here as a distinction to the previous "prototype" HANNA, but in what way does the new method enable arbitrary components that the original HANNA could not model? Please explain.

Miscellaneous

- Page 2: UNIFAC as standard gE models across industry and academia -> Do the authors mean prediction model for phase equilibria? Otherwise that is a reach with NRTL or other gE models and equations of state being in the mix as well.

- Page 2: The paragraph on COSMO needs some revision: COSMO is a model for solvation effects, what the authors are trying to describe is COSMO-RS/COSMO-SAC. "COSMO methods from quantum chemistry" is not quite descriptive, the quantum chemistry method that underlies the surface charge density calculation is DFT.

- Page 3: "Strictly complies with all relevant physical laws and constraints" -> which are? Gibbs-Duhem-consistency and limiting behavior? Please be more explicit. (Same also for the caption of Figure 1: "all thermodynamic constraints")

- Page 7: The authors keep referring to NRTL and UNIQUAC as lattice models which is only true for the latter. They also overstate the physical basis of those common gE models. While there were plenty of considerations of physics going into the development of these models, we can also acknowledge that in chemical engineering practice those models are used as purely empirical models, where the "pair-interaction" parameters bear no physical meaning. The authors might want to look towards molecular equations of state for a comparison with an actual physics-based extrapolation to mixture.

- Page 7: "basically any system" -> which limitation is implied with "basically"?
- Page 7: "and for phase equilibria with supercritical components (using Henry's law) or SLE" -> Like with any other gE model that requires additional data/parameters. Unclear why this would be an advantage of HANNA. Please clarify.
- Page 7: "the Poynting correction should be taken into account" -> Yes, and the non-ideality of vapor phase. While it is generally reasonable to exclusively position the proposed method in comparison to gE models, this would be the place where the authors should mention equations of state as powerful and ubiquitous models for phase equilibria (rather than the Poynting correction).
- Page 7: "weak electrolytes" -> unclear. HANNA was trained on binary data, so how do weak electrolyte systems fit into that picture. Please elaborate.
- Page 8: Is the limitation regarding the tokenization somehow confined to CO? Intuitively this could also affect similar molecules like C=O or CCO? Could you elaborate?
- Page 8: "we have overcome limitations of conventional phase equilibrium solvers" -> What limitations are the authors referring to? Robust methods for T_p flash calculations have been available for decades.
- Page 9: "extend the temperature range of the training data at its lower end" -> That seems like conjecture. I would expect most experimental solubility data to be at room temperature, please check or provide some reference.
- Page 9: "and may be resolved by a workaround within ChemBERTa-2" -> unclear. Please elaborate.
- Page 9: "PyTorch's default settings" -> please add settings to the SI if they are relevant for reproduction.

(Remarks on code availability)

With the instructions in the README, I was able to run and modify the very clear example code without any obstructions.

Version 1:

Reviewer comments:

Reviewer #1

(Remarks to the Author)

My concerns have been addressed in the new version of the manuscript.

(Remarks on code availability)

Reviewer #2

(Remarks to the Author)

The authors have thoroughly replied to all the concerns raised in the initial reviews. In particular, I appreciate the comparison to literature models which provides additional context regarding the interpretation of error metrics (i.e., the typical range that similar models achieve) and through that even highlights the accuracy of the proposed model even more. I am happy to recommend the updated manuscript for publication in Nature Communications.

(Remarks on code availability)

NCOMMS-25-89872-T

General remarks and changes to the manuscript

In addition to the changes to our manuscript based on the comments of the reviewers, detailed below, we have made several additional improvements:

- We have updated our dataset to the latest (2025) version of the Dortmund Data Bank and retrained our model on this new data.
- In this process, we have significantly expanded the liquid-liquid equilibrium (LLE) database and now include experimental data points for which only the composition of one phase in equilibrium is documented.
- We have also extended the dataset with vapor-liquid equilibrium (VLE) data for which only the liquid-phase composition, but not the vapor-phase composition, is reported (TPX data).
- Additionally, we now include excess enthalpy (HE) data in the training and evaluation of our model HANNA.
- In the Supporting Information, we have added a detailed evaluation for the new data types (TPX and HE), demonstrating HANNA's very good accuracy also for predicting these data.

Responses to reviewer's comments

Times: reviews (full text)

Courier: responses (page numbers given in the responses refer to the "Marked Revision")

Blue marked: Changes to the text in the manuscript

Reviewer #1

In the present manuscript a generalization of a machine learning (ML) based gE model, called HANNA, is presented. This tool could be used for the prediction of VLE and LLE of binary and multicomponent systems only from the molecular structure of the components, avoiding a model for gE or the activity coefficient.

The manuscript addresses a topic of great relevance and offers valuable insights that could guide future research in this field. The data analysis, interpretation, and conclusions are sound. In my opinion, this paper deserves to be published and will have a remarkable impact in this research area.

We thank the reviewer for their positive and valuable feedback.

Below just some comments:

One of the weak points of the method is the extrapolation from binary to multicomponent systems using the geometric projection of the binary gE . Also, not having used ternary and quaternary data for the training, drastically could reduce the possibility of obtaining reliable results for many multicomponent systems. For engineering calculations, the qualitative description of phase equilibrium behaviour is usually not enough (not even a rough quantitative one), as for example the island type LLE prediction presented in Figure 2c. For equipment and process design high accuracy is required. This poses a great challenge for multicomponent systems with different types of equilibrium regions involving vapor, liquid and anhydrous or hydrated solid phases (VLE, LLE, VLLE, LLSE, LLShSa,...). If “special” components are considered as polymers, ionic liquids, etc... the complexity could be even higher.

We agree with the reviewer that, for the final process design, highly accurate descriptions of thermodynamic properties are required, which are usually only possible via dedicated models tailored to individual systems or via experiments. The goal of this work is different. It is to provide a broadly applicable model that enables reliable predictions for the early stages of process design. The method we introduce has a significantly broader applicability and often better prediction accuracy for binary and multi-component mixtures than the best-established methods (UNIFAC and other benchmark ML models, cf. below).

We furthermore demonstrate in Figs. S.15 and S.16 in the Supporting Information that the extrapolation from binary to multi-component mixtures that we use in the HANNA framework is comparable or better than that of the most successful models of the excess Gibbs energy, NRTL and UNIQUAC, even after fitting them to data for all binary subsystems.

However, we agree with the reviewer that further improvements could be achieved by refining the geometric projection method, incorporating multi-component data into HANNA training, or introducing ternary (or even higher-order) parameters in future work, as we discuss in the Conclusions.

The paper does not indicate clearly the type of systems that can be addressed by HANNA. I have tested the interactive interface MLPROP, provided by the authors, to run some checks. For example, ternary type III (Treybal classification) with three liquid phases in equilibrium (LLLE tie-triangle) and island type ternary systems at constant temperature can be predicted (at least qualitatively). However, it is truth that the results in some cases could be considerably deviated from the experimental ones (source DECHEMA Chemistry Data Series). For example, I have found discrepancies in the following arbitrarily chosen systems:

- a) 1-Hexanol(1) + Nitromethane(2) + Water(3) at T=294K (type III with LLLE)
- b) acetic acid (1) + dimethyl formamide (2) + cyclohexene (3) 291.15K (island)
- c) dichloroacetic acid (1) + 1-methyl-2-pyrrolidinone (2) + 1-hexene (3) 293.15 (island)
- d) dichloroacetic acid (1) + N-formylpiperidine (2) + cyclohexane (3) 298.15K (island)

The goal of this work is very ambitious, and so one can imagine that incorrect results will be obtained for some (or even many!) predictions of multicomponent systems at this stage of the research. Along with the interface MLPROP the authors indicate (with very good judgment): “no warranty is given that the provided tool here is error-free, or that the results obtained with the tool are correct”. Regardless of this, the relevant is that this study identifies a promising direction for further investigation in phase equilibria.

We thank the reviewer for this comment and agree: as our method is broadly applicable, there will be instances, where its predictions will not be accurate. To further illustrate this, we have added Fig. S.5 to the Supporting Information, which shows predictions for type III and island-type LLE in ternary systems, including two of the systems proposed by the reviewer. On page 8 in the manuscript, we now refer to Fig. S5 as follows:

While the results in Fig. 3c cover ternary LLE with one (top) and two (bottom) binary subsystems that exhibit a miscibility gap, LLE examples for systems with complete miscibility of the binary subsystems (island-type LLE behavior) and miscibility gaps in all three binary subsystems (leading to a three-phase LLLE) are given in Fig.~S.5 in the Supplementary Information. HANNA is also able to predict these correctly, at least qualitatively.

Reviewer #2 (Remarks to the Author):

The authors present an update to their prediction model for activity coefficients (HANNA). The new model is trained on VLE and LLE data and can be evaluated for multicomponent mixtures using a simple but effective geometric projection method that provides physically meaningful results on the binary edges of the composition space. Overall this is an important and novel contribution that tackles limitations of existing models. However, I see some shortcomings that the authors should address before publication.

We thank the reviewer for their thorough review and feedback to our manuscript. We have addressed all points as discussed in detail in the following.

Overall capability and impact of the proposed method

- The title of the manuscript mentions an expression for the excess Gibbs energy, which the manuscript never delivers. Of course, it is possible to write down an expression for any neural network in theory, but in practice that is cumbersome. I invite the authors to rethink the expectations they set with the title.

We thank the reviewer for the advice and have changed the title to:

"Thermodynamically consistent machine learning model for excess Gibbs energy"

- While the authors do not spare superlatives ("excellent predictions", "clearly outperforming", "superior performance", "entire relevant physical performance") the presented results are not fully backing those claims: The MAE is slightly smaller compared to UNIFAC ("clearly outperforming") and worsens again somewhat when evaluated on systems that UNIFAC can't model ("retains its accuracy").

We thank the reviewer for this feedback and have carefully revised the manuscript to remove statements that might be perceived as promotional.

For ACI the predictions are way better, but UNIFAC was not developed for that and should (at least responsibly) not be used for it.

We respectfully disagree here. UNIFAC was developed to predict activity coefficients over the entire composition range, including dilute solutions. Its application to dilute components and limiting activity coefficients is therefore fully consistent with its original scope. We are in close contact with the developers of UNIFAC, who likewise consider its use at high dilution appropriate.

The authors keep stating that UNIFAC fails on LLEs, but in their own results (based on the MAE) UNIFAC seems to work reasonably well for LLE?

We thank the reviewer for this comment. We have revised and moderated our statements regarding the performance of UNIFAC for LLE.

OF course the comparison is flawed, because many of the systems used in the evaluation were used in the determination of UNIFAC parameters, but on the other hand, UNIFAC is a much simpler model. Why not compare it to different ML models instead or as well? Wouldn't it be interesting to see how it compares to soft-constrained models or ones that are not trained on LLE data? The authors should dial back their claims or back them up with meaningful data.

We fully agree and have made additional comparisons with other ML-based models from the literature. The results are described in a new subsection "Comparison to other machine learning models", in which we benchmark HANNA against the four best alternative models that rely on ML algorithms combined with physical knowledge. The results demonstrate that HANNA has higher accuracy than all benchmark models in both prediction accuracy and application range. The new section, including Table 1 comparing the scores of all models, can be found on pages 10-12 in the revised manuscript.

In the following, we also compare HANNA against different ML models from the literature. We thereby consider only ML models that, like HANNA, are not based on the equation framework of a physics-based gE model, require only the SMILES of the components as input, and for which the model parameters are disclosed or can be reproduced with the available code. Specifically, we have selected GE-GNN~\cite{Rittig2024}, GDI-GNN~\cite{Rittig2023}, SolvGNN~\cite{Qin2023}, and GNN-IAC~\cite{SanchezMedina2022} as additional benchmarks. Compared to HANNA, all these models have limitations in their applicability: they are only applicable to binary systems (GE-GNN, GDI-GNN, GNN-IAC), to the state of infinite dilution (GNN-IAC), or to predictions at T = 298 K (all). Furthermore, they lack strict consistency with the Gibbs-Duhem equation (GDI-GNN, SolvGNN) or in the limiting case of pure components (GE-GNN, GDI-GNN, SolvGNN), whereas HANNA complies with all relevant constraints. Tab.~S.2 in the Supplementary Information summarizes the ranges of applicability of these models and the consistency criteria they comply with.

For a fair comparison, we evaluate HANNA and the other ML models on common subsets of our data for which predictions are feasible with all models. While the results of HANNA on these subsets are true predictions (since the respective systems were not part of the training or validation sets), we cannot rule out that some of them were included in the training sets of the literature models. We note that GE-GNN, GDI-GNN, and SolvGNN have not been trained on experimental data but on synthetic data generated with COSMO-RS~\cite{Klamt1995}.

The results of the comparison are summarized in Tab. 1. The scores of GE-GNN, GDI-GNN, and SolvGNN on the binary TPXY, TPX, and ACI data are almost identical, which could be expected, as all three models rely on the same GNN architecture and training data set and only differ in the

final step of the prediction of the activity coefficients and/or the regularization during training. In fact, the authors arrived at a similar conclusion when comparing GE-GNN against GDI-GNN and SolvGNN on their extrapolation test data set~\cite{Rittig2024}. For the TPXY, TPX, and ACI data, HANNA shows significantly lower scores (higher accuracy) for both binary and ternary systems. On the LLE subset, HANNA achieves the lowest mean MAE_{sys}, suggesting the fewest outliers, while no significant difference in the median MAE_{sys} between the models is observed. Overall, HANNA shows the best performance, with the highest prediction accuracy in almost all cases and a significantly broader applicability than all benchmark models, cf. Tab.~S.2 in the Supplementary Information.

Table 1: Comparison of HANNA with different ML models from the literature. We evaluate all models (including HANNA) only on isothermal data at 298 K and exclude all systems containing ionic liquids from the comparison. Because the architecture of GNN-IAC is not suited to processing water, systems that contain water are omitted from the binary ACI comparison. The mean and median of the MAE_{sys} for each data type are given. The best scores for each data type are bold. The results for HANNA are for test data; the other models were used as published.

Model	TPXY		TPX		ACI		LLE		
	Mean	Median	Mean	Median	Mean	Median	Mean	Median	
Binary	HANNA	0.165	0.066	0.151	0.050	0.229	0.141	0.032	0.011
	GE-GNN	0.227	0.108	0.272	0.091	0.438	0.296	0.051	0.012
	GDI-GNN	0.225	0.109	0.278	0.090	0.433	0.288	0.046	0.009
	SolvGNN	0.225	0.109	0.280	0.091	0.432	0.293	₃	₃
	GNN-IAC	₄	₄	₄	₄	0.372	0.150	₄	₄
Ternary	HANNA	0.646	0.143	0.310	0.141	0.429	0.257	–	–
	SolvGNN	0.689	0.189	0.667	0.328	0.841	0.758	₃	₃

³Due to large oscillations in the predicted activity coefficients over the concentration, SolvGNN could not be evaluated on LLE data.

⁴GNN-IAC can only predict activity coefficients at infinite dilution.

- Page 5: How are the examples in Fig. 2b-d chosen (and Fig. 3b-c)? Without any indication on the decision process or at least an indication of the MAE of the systems, how are the readers not supposed to think that those are cherrypicked for the best agreement with experimental data? Please elaborate.

We agree and have added the error scores for each example shown in the plots, demonstrating that the chosen examples are representative.

- Page 8: "It will improve process design and optimization, and ease their application" -> This ties in with the point above regarding the title. The model itself is fine as it is, but the authors don't describe how it would be used in design or optimization. Are you suggesting that simulation software developers will include the NN architecture of HANNA into their

products? Or is it not more realistic that process engineers will fit their favorite gE model to the HANNA outputs in which case the advantage of the additional flexibility is obsolete. Please elaborate.

We thank the reviewer for pointing at this missing statement and have made the following changes on page 14 to clarify this:

Because we release both the full model architecture and trained parameters as open source, process engineers can integrate HANNA directly into existing workflows and simulation software. Users who prefer to continue to use conventional thermodynamic models can, e.g., fit NRTL or UNIQUAC parameters to HANNA's predictions. These fitted parameters can directly be used in established simulation frameworks without additional implementation effort, albeit typically with some loss in accuracy relative to the original HANNA predictions. A routine for fitting NRTL parameters to HANNA is also provided in MLPROP.

Literature context

- Page 2: The literature discussion at the bottom of page 2 leaves out relevant work on the prediction of VLEs with activity coefficient/gE models. What about the work on GNNs by Zavala and coauthors, soft-constrained models like the "Gibbs-Duhem-informed NN" by Rittig et al. (<https://doi.org/10.1039/D3DD00103B>), potentially Medina et al. on prediction of infinite dilution activity coefficients.

We agree and have added a paragraph discussing these models and their limitations compared to HANNA to the Introduction on page 3:

Furthermore, several ML models for the prediction of activity coefficients have been proposed that are not based on the equations of any of the classical gE models. Medina et al. presented graph neural network (GNN) architectures for the prediction of infinite-dilution activity coefficients at constant temperature (GNN-IAC)~\cite{SanchezMedina2022}, and with temperature dependence (GH-GNN)~\cite{SanchezMedina2023}, using the Gibbs-Helmholtz equation. Qin et al. \cite{Qin2023} have developed a GNN (SolvGNN) for the prediction of composition-dependent, isothermal activity coefficients in binary and ternary mixtures. Based on the SolvGNN architecture, Rittig et al. have built the Gibbs-Duhem-informed GNN (GDI-GNN) \cite{Rittig2023} and the excess Gibbs free energy GNN (GE-

GNN) \cite{Rittig2024}, which include soft-constraints and hard-constraints on the Gibbs-Duhem equation, respectively.

- Page 3: "The first ML-based gE-model ..." -> I suppose that might have been true at the publication of the HANNA "prototype" but by now other work has been published like <https://doi.org/10.1039/D4SC04554H>. The presented methodology reads as an extension of HANNA to include LLE predictions and multicomponent systems, which is certainly novel and exciting, but the paragraph is not clear. Please rephrase to make the reader more aware of (1) what has been published previously by the authors, (2) what other developments have been done in the meantime, (3) what is the novelty presented in this publication.

We agree and have added the following explanation of the original HANNA model and the consistency criteria right after the extended discussion of the ML literature context above to the manuscript on page 3-4:

In recent work, we have introduced HANNA~\cite{Specht2024}, a model that combines the flexibility of artificial neural networks with explicit physical knowledge hard-coded into the network architecture. In its first version, HANNA was trained on binary activity coefficients derived from experimental VLE data, as well as experimental activity coefficients at infinite dilution (ACI). HANNA is the first model to predict temperature- and composition-dependent activity coefficients without being based on an underlying physics-based gE model, while still strictly complying with all relevant thermodynamic consistency criteria. [...]

In this work, a generalization of HANNA is presented that is flexible enough for the holistic description of different phase equilibria (namely VLE and LLE), enables predictions for multi-component mixtures comprising any number of components, and thereby still strictly complies with all thermodynamic constraints. To achieve this, we significantly improve the training process, which now additionally includes end-to-end training on experimental LLE and excess enthalpy data. Moreover, we implement a geometric projection method for extrapolation to any multi-component mixture.

- Page 3: "enables prediction for arbitrary components" -> This is used here as a distinction to the previous "prototype" HANNA, but in what way does the new method enable arbitrary components that the original HANNA could not model? Please explain.

We understand that the wording might lead to misunderstandings and have changed the respective sentence on page 4 of the manuscript to (see also comment above):

In this work, a generalization of HANNA is presented that is flexible enough for the holistic description of different phase equilibria (namely VLE and LLE), enables predictions for multi-component systems comprising any number of components, and thereby still strictly complies with all thermodynamic constraints.

Miscellaneous

- Page 2: UNIFAC as standard gE models across industry and academia -> Do the authors mean prediction model for phase equilibria? Otherwise that is a reach with NRTL or other gE models and equations of state being in the mix as well.

Indeed, we here refer to models from the UNIFAC family as the standard gE-model for the predictive modeling of phase equilibria, when no data for fitting other gE models are available. We have updated the sentence on page 3 accordingly and now write:

Despite the described shortcomings, the different UNIFAC variants have long been and remain the standard gE models used for the predictive modeling of phase equilibria across industry and academia, when no data for fitting parameters for models such as UNIQUAC and NRTL are available.

- Page 2: The paragraph on COSMO needs some revision: COSMO is a model for solvation effects, what the authors are trying to describe is COSMO-RS/COSMO-SAC. "COSMO methods from quantum chemistry" is not quite descriptive, the quantum chemistry method that underlies the surface charge density calculation is DFT.

We thank the reviewer for pointing out the imprecise formulation regarding COSMO and the related models. We have revised the respective paragraph and now write on page 3 of the manuscript:

An alternative to the GC models of the UNIFAC family are the gE models based on quantum-chemical DFT calculations combined with the conductor-like screening model (COSMO)~\cite{Klamt1993}. Different variants of such frameworks have been published, most notably COSMO-RS~\cite{Klamt1995} and COSMO-SAC~\cite{Lin2001}. While these frameworks avoid group-interaction parameter tables and can, in principle, be applied to any system, they often require expensive calculations, and the results can depend on the chosen DFT level and COSMO parametrization~\cite{Grensemann2005, Wang2005, Mu2007,

Franke2011}. For systems where both approaches can be applied, UNIFAC is often reported to be more accurate than COSMO frameworks~\cite{Grensemann2005, Fingerhut2017,Xue2012}.

- Page 3: "Strictly complies with all relevant physical laws and constraints" -> which are? Gibbs-Duhem-consistency and limiting behavior? Please be more explicit. (Same also for the caption of Figure 1: "all thermodynamic constraints")

We have added a comprehensive description of all relevant thermodynamic constraints on page 3-4:

These comprise the consistency of the activity coefficients with the Gibbs-Duhem equation, permutation-equivariance (a permutation in the component input order must be equally reflected in the order of the activity coefficients in the output; their values must not change), their correct behavior at infinite dilution (for $x_i \rightarrow 1$, $\gamma_i \rightarrow 1$), and consistency for pseudo-mixtures (identical components in the mixture must reduce to one pure component).

- Page 7: The authors keep referring to NRTL and UNIQUAC as lattice models which is only true for the latter. They also overstate the physical basis of those common gE models. While there were plenty of considerations of physics going into the development of these models, we can also acknowledge that in chemical engineering practice those models are used as purely empirical models, where the "pair-interaction" parameters bear no physical meaning. The authors might want to look towards molecular equations of state for a comparison with an actual physics-based extrapolation to mixture.

We have replaced the term 'lattice model' on page 2 of the manuscript with the term 'model based on the local-composition theory', which is true for both NRTL and UNIQUAC/UNIFAC.

Furthermore, we agree that all established thermodynamic models are used empirically, by fitting parameters to suitable data. We still find it important to distinguish between ML-based models that rely on highly flexible architectures and physics-based models such as NRTL and UNIQUAC, whose underlying equations are derived from physical theories and considerations, but we have revised our formulations to "physics-based models" at different places in the manuscript to avoid confusion.

- Page 7: "basically any system" -> which limitation is implied with "basically"?

We thank the reviewer for pointing out this imprecise formulation. We discuss the limitations regarding the

applicability in the Section "Limitations" in the results section. To avoid confusion we revised the concerning sentence in the manuscript on page 10 as follows:

In addition to its strong predictive performance, HANNA offers a much broader applicability than mod. UNIFAC, because it can, in principle, be applied to any mixture for which the molecular structures of the components are known; remaining limitations are discussed in the section "Limitations" below.

- Page 7: "and for phase equilibria with supercritical components (using Henry's law) or SLE" -> Like with any other gE model that requires additional data/parameters. Unclear why this would be an advantage of HANNA. Please clarify.

The cited sentence is from the "Limitations" section and states that, while in principle possible with HANNA, its accuracy for these tasks is yet to be tested. However, to avoid any confusion, we have reformulated the sentence and now write on page 9 of the manuscript:

Similar to other gE models, HANNA can also be used to predict activity coefficients for phase equilibria with supercritical components (using Henry's law) or solid-liquid equilibria (SLE).

- Page 7: "the Poynting correction should be taken into account" -> Yes, and the non-ideality of vapor phase. While it is generally reasonable to exclusively position the proposed method in comparison to gE models, this would be the place where the authors should mention equations of state as powerful and ubiquitous models for phase equilibria (rather than the Poynting correction).

We discuss modeling activity coefficients in the liquid phase here but agree that mentioning the Poynting correction may cause confusion. We have omitted this rather than starting a textbook-level discussion on how to model vapor-liquid equilibria at high pressures. We simply write now on page 12:

Furthermore, HANNA, like most gE models, neglects the pressure dependence of the liquid phase activity coefficients. While this is a valid assumption for most engineering applications, it may break down at very high pressures.

- Page 7: "weak electrolytes" -> unclear. HANNA was trained on binary data, so how do weak electrolyte systems fit into that picture. Please elaborate.

Our training data set includes components that are weak electrolytes (substances that tend to dissociate partially,

such as acetic acid in water). These are treated as regular molecular species within HANNA. To clarify this in the manuscript, we have added the following sentence to page 12:

Weak electrolytes that may dissociate partially and ionic liquids were also included in the training set and are therefore within the current scope of HANNA. However, HANNA does not treat ions explicitly. Instead, any effects of dissociation are implicitly embedded in the binary training data. The extension to explicit modeling of the ionic species is beyond the scope of this work.

- Page 8: Is the limitation regarding the tokenization somehow confined to CO? Intuitively this could also affect similar molecules like C=O or CCO? Could you elaborate?

The reviewer raises a very interesting point here. Based on the tokenization, other molecules could also be affected. For this reason, in our extended data set used during the revision, we systematically checked all LLE predictions for the peculiar phase behavior. Indeed, aside from methanol (SMILES: CO), we also found a few cases in which systems containing water (SMILES: O) and ethanol (SMILES: CCO) were affected. Interestingly, despite the completely different architecture and training data, we observed a similar behavior when evaluating the literature model GE-GNN, with some systems containing water and methanol also affected. Furthermore, the same behavior was recently observed for LLE prediction with COSMO-SAC: <https://doi.org/10.1039/D5DD00259A>. To discuss these points, we have revised the manuscript on pages 12-13 as follows:

The only exception to this was a small fraction (<1%) of the binary LLE systems, for which we found that HANNA predicts the existence of two separate LLE phase regions at elevated temperatures. Although this is thermodynamically possible, it has, to our knowledge, not been experimentally observed in these systems. Most of the affected systems contain methanol, ethanol, or water. We speculate that the ChemBERTa-2~\cite{Ahmad2022} tokenization for methanol (SMILES: 'CO') and ethanol (SMILES: 'CCO') into 'C' (equivalent to the SMILES for methane) and 'O' (equivalent to the SMILES for water) results in embeddings that cannot fully capture the properties of these small molecules.

[..]

Despite the fundamentally different architecture and training data, we observe a similar behavior for predictions from GE-GNN, where some systems containing methanol or water are also

affected. Note that Antolović et al.~\cite{Antolovic2025} have recently encountered similar anomalies for predictions of LLE with COSMO-SAC.

- Page 8: "we have overcome limitations of conventional phase equilibrium solvers" -> What limitations are the authors referring to? Robust methods for T_p flash calculations have been available for decades.

We have changed the paragraph on page 14 of the manuscript and now elaborate specifically on the advantages of the CEM for our work:

Using the CEM to determine LLE, we have obtained phase-split predictions in a numerically robust and initialization-independent manner for thousands of systems in an automated workflow. Unlike conventional solvers, the CEM does not require careful initialization or phase-stability checks to identify the correct number of coexisting phases.

- Page 9: "extend the temperature range of the training data at its lower end" -> That seems like conjecture. I would expect most experimental solubility data to be at room temperature, please check or provide some reference.

Our statement concerning the temperature range is based on the data situation in the Dortmund Data Bank (DDB), which is also highlighted in the paper "A Modified UNIFAC (Dortmund) Model. 4. Revision and Extension":

<https://pubs.acs.org/doi/10.1021/ie0108043>. Fig. R1 below shows the temperature distributions for VLE, SLE, and LLE in the current version of the DDB, clearly indicating a larger number of SLE data points available at lower temperatures.

Figure R1 – Temperature distribution of the VLE, LLE, and SLE data in the DDB (version 2025).

In the manuscript, we now cite the reference stated above on page 14:

These data would extend the temperature range of the training data at its lower end, c.f. Ref. \cite{Gmehling2002}.

- Page 9: "and may be resolved by a workaround within ChemBERTa-2" -> unclear. Please elaborate.

We thank the reviewer for this comment. We now discuss an option on page 15 of the manuscript:

Specifically, one of the unused tokens in ChemBERTa-2 could be assigned to the water molecule, such that water and, e.g., hydroxyl groups no longer share the same token.

- Page 9: "PyTorch's default settings" -> please add settings to the SI if they are relevant for reproduction.

We thank the reviewer for pointing this out. The reviewer's comment refers to the sentence "In all cases, we use the Sigmoid Linear Unit (SiLU) activation function with PyTorch's default settings." in the Methods Section. Upon reviewing the PyTorch documentation again, we found that the SiLU function has no parameters that can be specified. Therefore, the phrase "with PyTorch's default settings" is obsolete, and we have removed it from the manuscript.

Reviewer #2 (Remarks on code availability):

With the instructions in the README, I was able to run and modify the very clear example code without any obstructions.

We thank the reviewer for testing our example code and are happy to hear that they were able to run it without any problems.